# Combinatorial expression of γ-protocadherins regulates synaptic connectivity in the mouse neocortex

Yi-jun Zhu[1,2,3], Cai-yun Deng[1], Liu Fan[2], Ya-Qian Wang[2], Hui Zhou[2], Hua-tai Xu[1,2]*

[1]Institute of Neuroscience and State Key Laboratory of Neuroscience, CAS Center for Excellence in Brain Science and Intelligence Technology, Chinese Academy of Sciences, Shanghai, China; [2]Lingang Laboratory, Shanghai Center for Brain Science and Brain-Inspired Intelligence Technology, Shanghai, China; [3]University of Chinese Academy of Sciences, Beijing, China

**Abstract** In the process of synaptic formation, neurons must not only adhere to specific principles when selecting synaptic partners but also possess mechanisms to avoid undesirable connections. Yet, the strategies employed to prevent unwarranted associations have remained largely unknown. In our study, we have identified the pivotal role of combinatorial clustered protocadherin gamma (γ-PCDH) expression in orchestrating synaptic connectivity in the mouse neocortex. Through 5' end single-cell sequencing, we unveiled the intricate combinatorial expression patterns of γ-PCDH variable isoforms within neocortical neurons. Furthermore, our whole-cell patch-clamp recordings demonstrated that as the similarity in this combinatorial pattern among neurons increased, their synaptic connectivity decreased. Our findings elucidate a sophisticated molecular mechanism governing the construction of neural networks in the mouse neocortex.

*For correspondence:
xuht@lglab.ac.cn

## eLife assessment

The authors used an innovative modified 10X genomic sequencing method to detect cPCDHg is-forms in pyramidal neurons. With **solid** electrophysiological recordings, they showed that neurons expressing the same sets of cPCDHg isoforms are less likely to form synapses with each other. These **valuable** findings confirms previous results and extend our understanding of cPCDHg diversity and neuronal connectivity.

## Introduction

The precision of synaptic connections is vital for the functioning of neural circuits (*Yogev and Shen, 2014*). Cell adhesion molecules play crucial roles in the specificity of synapse formation (*Duan et al., 2014*; *Serizawa et al., 2006*; *Tan et al., 2015*; *Rawson et al., 2017*; *Sytnyk et al., 2017*; *Berns et al., 2018*; *Jang et al., 2017*; *Courgeon and Desplan, 2019*). However, how to achieve such specificity at the microcircuit level remains an open question. The unique expression pattern of clustered protocadherins (cPCDH) leads to millions of possible combinations of cPCDH isoforms on the neuron surface (*Kaneko et al., 2006*; *Esumi et al., 2005*), effectively serving as a distinctive barcode for each neuron (*Yagi, 2012*). Notably, the absence of γ-PCDH does not induce general abnormalities in the development of the cerebral cortex, including cell differentiation, migration, and survival (*Wang et al., 2002*; *Garrett et al., 2012*). However, γ-PCDH presence has been detected at synaptic contacts (*Fernández-Monreal et al., 2009*; *Phillips et al., 2003*; *LaMassa et al., 2021*), and its absence has substantial effects on neuronal connectivity (*Tarusawa et al., 2016*; *Kostadinov and Sanes, 2015*; *Lv*

*et al., 2022*). While the homophilic properties of γ-PCDH promote dendritic complexity (*Molumby et al., 2016*), emerging evidence suggests that it might hinder synapse formation. Previous studies indicate that homophilic interactions, facilitated by large overlapping patterns of cPCDH isoforms on opposing cell surfaces, may lead to intercellular repulsion (*Rubinstein et al., 2015*; *Brasch et al., 2019*; *Honig and Shapiro, 2020*; *Lefebvre et al., 2012*). Consistent with the repulsion concept, the absence of γ-PCDH results in significantly more dendritic spines and inhibitory synapse densities in neocortical neurons (*Molumby et al., 2017*; *Steffen et al., 2021*). Paralleling this, neurons overexpressing one of the γ-PCDH isoforms exhibit significantly fewer dendritic spines (*Molumby et al., 2017*). Furthermore, the absence of the clustered PCDH augments local reciprocal neural connection between lineage-related neurons in the neocortex (*Tarusawa et al., 2016*; *Lv et al., 2022*), even when sister cells exhibit more similar expression patterns of γ-PCDH isoforms (*Lv et al., 2022*).

Intriguingly, while γ-PCDH appears to have 'contradictory' effects on dendritic complexity and dendritic spines, it negatively influences synapse formation in the forebrain. It is important to note that each neuron expresses multiple isoforms of γ-PCDH (*Kaneko et al., 2006*; *Lv et al., 2022*). What is the impact of this combinatorial expression on synapse formation? In this study, using 5' end single-cell sequencing, we revealed the diversified combinatorial expression of γ-PCDH isoforms in neocortical neurons. Through multiple whole-cell patch-clamp recordings after the sequential in utero electroporation, we discovered that the combinatorial expression of γ-PCDHs empowers neurons to decide which partners to refrain from forming synapses with, rather than merely determining which ones to engage in synaptogenesis with.

## Results

### The diversified combinatorial expression pattern of γ-PCDHs in neocortical neurons revealed by 5' end single-cell sequencing

The gamma cluster of cPCDHs (γ-PCDHs) is critical for synaptic connectivity (*Kostadinov and Sanes, 2015*; *Tarusawa et al., 2016*; *Lv et al., 2022*; *Weiner et al., 2005*). To determine the role of γ-PCDHs in the neocortex, we examined their expression in the neocortical neurons of postnatal mice. Existing research has suggested that cPCDHs are expressed stochastically in Purkinje cells and olfactory sensory neurons (*Hirayama et al., 2012*; *Toyoda et al., 2014*; *Mountoufaris et al., 2017*). In this study, we harnessed the power of 5' end single-cell RNA sequencing to precisely identify γ-PCDH isoforms by focusing on their variable exon, exon 1, where they differ from each other (*Kohmura et al., 1998*; *Wu and Maniatis, 1999*). Given that the second postnatal week is a critical stage for synapse formation in the rodent neocortex (*Lendvai et al., 2000*; *Holtmaat and Svoboda, 2009*), we chose postnatal day 11 (P11) as the time point for our examination. Following reverse transcription and cDNA amplification (*Figure 1—figure supplement 1A and B*), we divided the cDNA into two segments: one designed for the specific amplification of *Pcdhg* mRNAs and the other for the construction of a 5' gene expression library (*Figure 1—figure supplement 1C*). After the cluster analysis (*Figure 1—figure supplements 2–4*), we collected 6505 neurons from an initial pool of 17,438 cells (*Figure 1A*, *Figure 1—figure supplement 1D*). For in-depth analysis, we focused on neurons expressing more than 10 unique molecular identifiers (UMIs) for all γ-PCDH isoforms (cutoff >1 for each type of individual isoform) (*Figure 1B and C*). We observed the near-ubiquitous expression of 'C-type' isoforms, specifically C3, C4, and C5 (*Figure 1D*). It is important to note that the fraction of cells expressing 'C-type' isoforms was significantly higher when compared to 'variable' isoforms (*Figure 1D*, *Figure 1—figure supplement 1E*), which is consistent with findings from a previous study (*Toyoda et al., 2014*).

We proceeded to conduct a pairwise analysis to assess the similarity of 'variable' isoforms among neurons regarding the single-cell expression pattern of γ-PCDHs. This extensive analysis revealed that the majority of neocortical neurons from all clusters exhibited very low similarity level (*Figure 1E*, *Figure 1—figure supplements 5 and 6*). This finding strongly suggests distinct combinatorial expression patterns among these neurons. Given that all our electrophysiological recordings were carried out on pyramidal neurons in layer 2/3 of the neocortex, we conducted more detailed analysis, including the examination of detailed expression of γ-PCDHs in individual neurons, specifically in the corresponding cluster 7. All these analyses consistently revealed a diverse range of combinatorial expression patterns among neurons in this cluster, a phenomenon in alignment with the general population of neocortical neurons (*Figure 1—figure supplement 5*).

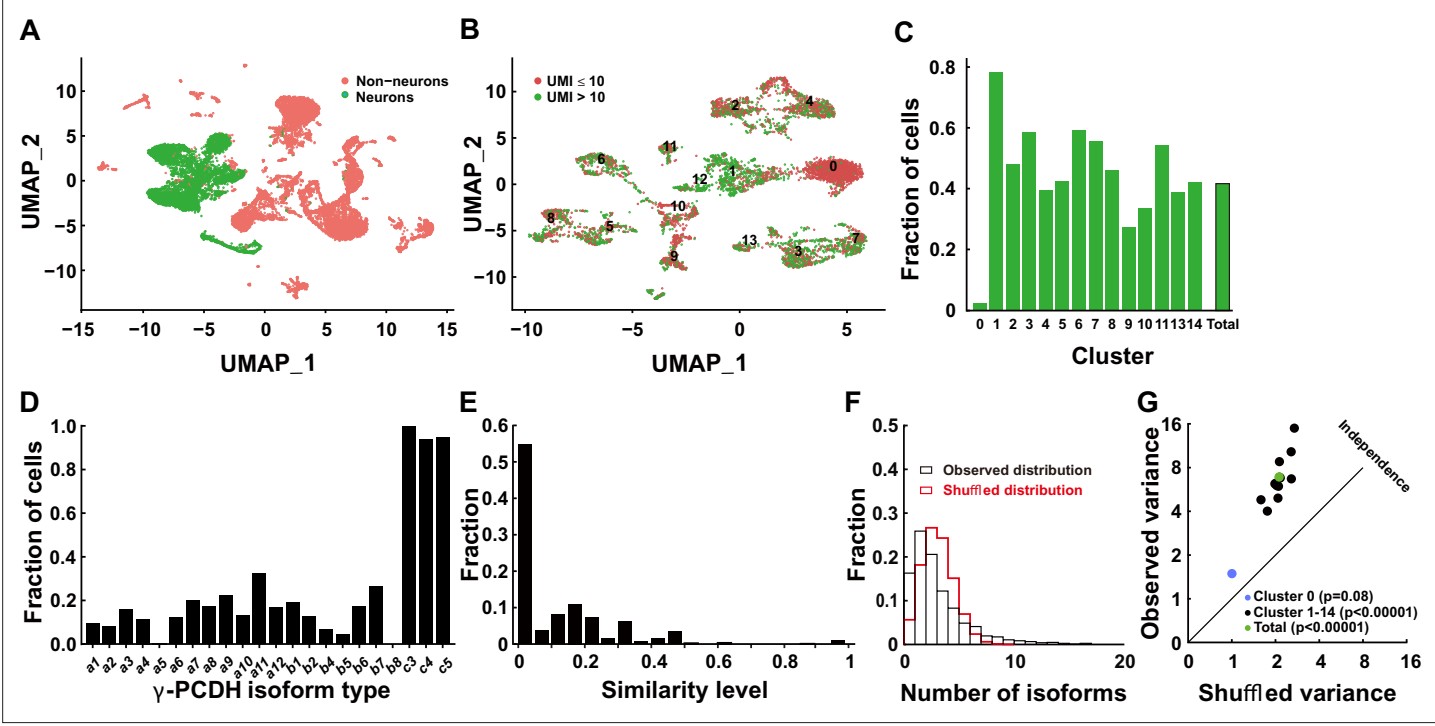

**Figure 1.** Diversified expression of *Pcdhg* isoforms in neocortical neurons. (**A**) Uniform Manifold Approximation and Projection (UMAP) analysis displaying 17,438 cells obtained through 5' end single-cell sequencing after data cleanup and doublet removal. Neurons are depicted as green dots, while non-neuronal cells are marked in red. (**B**) UMAP clusters of all neurons categorized by the unique molecular identifier (UMI) cutoff. Red dots denote cells with fewer than 10 total UMIs of *Pcdhg* (n = 3671), and green dots denote cells with more than 10 UMIs (n = 2834). (**C**) Fractions of neurons with more than 10 UMIs in different clusters. (**D**) Distribution of neurons expressing different *Pcdhg* isoforms in the neocortex. (**E**) Fraction distribution illustrating similarity levels in the combinatorial expression of *Pcdhg* variable isoforms among neurons. Similarity levels were calculated as $\frac{A \cap B}{A \cup B}$. (**F**) Observed distribution (black) of the fraction of cells expressing varying numbers of isoforms across all neurons. The red curve represents the shuffled distribution generated under the hypothesis of stochastic isoform expression. (**G**) The variance difference between the observed and shuffled fraction distribution of cells from all clusters. Source data is available on Dryad (https://doi.org/10.5061/dryad.931zcrjm7) and in *Figure 1—source data 1*.

The online version of this article includes the following source data, source code, and figure supplement(s) for figure 1:

**Source data 1.** Raw files of Cell Ranger output.

**Source code 1.** Source codes for data processing.

**Figure supplement 1.** Overview of the 5' end single-cell sequencing procedure for *pcdhg* isoforms.

**Figure supplement 2.** Single-cell RNA sequencing data profiling.

**Figure supplement 3.** Neuron selection.

**Figure supplement 4.** Neuron clusters.

**Figure supplement 5.** Diversely combinatorial expression of γ-PCDH in L2/3 neurons of cluster 7.

**Figure supplement 6.** Distribution of similarity levels among cell pairs across different clusters.

**Figure supplement 7.** A weak but significant co-occurrence of γ-PCDH variable isoforms in most of neocortical neurons (except neurons in cluster 0).

To further characterize this diversity, we employed variance analysis as per *Wada et al., 2018* to examine the distribution of the number of expressed isoforms per cell across all neurons (*Figure 1F*, *Figure 1—figure supplement 7*). As per Wada's definition of 'co-occurrence' (*Wada et al., 2018*), this primarily indicates potential interactions among different isoform expressions at the population level. This analysis uncovered a subtle but significant co-occurrence of γ-PCDH isoform expression in most neurons (*Figure 1G*). Notably, we did not detect any discernible differences among clusters, except for cluster 0, which displayed considerably lower expression (*Figure 1C*) and no co-occurrence of γ-PCDH isoforms (*Figure 1G*, *Figure 1—figure supplement 7*). In summary, the data derived from 5' end single-cell sequencing underscore the diverse and complex combinatorial expression of γ-PCDHs within the majority of neocortical neurons.

## The absence of γ-PCDHs increases local synaptic connectivity among pyramidal neurons in the neocortex

To delve into the function of γ-PCDH in the synaptic formation among neocortical neurons, we conducted paired recordings on pyramidal neurons in layer 2/3 of the neocortex from *Pcdhg* conditional knockout (cKO) mice. These genetically engineered mice were created by crossing *Pcdhg* $^{fcon3/fcon3}$ mice (*Lefebvre et al., 2008*; *Prasad et al., 2008*) with *Neurod6-cre* mice (*Goebbels et al., 2006*). This genetic combination specifically removed all variable and C-type γ-PCDH isoforms in pyramidal neurons. Our experimental setup involved multiple whole-cell patch-clamp recordings from cortical slices harvested from P9-32 mice, allowing us to measure the connectivity among nearby pyramidal cells (<200 μm between cell somas) in layer 2/3 of the neocortex by assessing the presence of evoked monosynaptic responses (*Figure 2*, *Figure 2—figure supplement 1*).

In the sample traces featured in the connectivity matrix obtained from six recorded neurons (*Figure 2A*, *Figure 2—figure supplement 1*), we observed two neuronal pairs (neuronal pairs 4→3 and 5→6) exhibiting unidirectional monosynaptic connections (indicated by orange arrows), and one pair (neuronal pair 1↔3) displaying bidirectional connections (indicated by green arrows) out of 15 possible pairings. Overall, our analysis revealed that the percentage of connected pairs was notably higher in *Pcdhg* cKO mice (20.2%, 103/511) than in wild-type (WT) mice (15.0%, 122/813) (*Figure 2B*). In light of the distinct roles that vertical vs. horizontal axes might play in synaptic organization within the neocortex (*Douglas and Martin, 2004*), we conducted an additional set of recordings on P10-20 mice, segregating neuron pairs along these axes. In *Pcdhg* cKO mice, we observed a more significant difference in connectivity for vertically aligned cells (18.3%, 94/515 in *Pcdhg* cKO mice vs. 11.2%, 73/651 in WT mice for vertically aligned neuron pairs; 12.2%, 27/221 in *Pcdhg* cKO mice vs. 9.5%, 28/294 in WT mice for horizontally aligned neuron pairs) (*Figure 2C*). We also created *Pcdha* cKO mice (*Figure 2—figure supplements 2–4*) and carried out similar experiments focused on vertically aligned neurons. In these mice lacking α-PCDH, we did not observe a significant difference (11.3%, 38/337 in *Pcdha* cKO mice vs. 14.3%, 26/182 in WT mice, *Figure 2D*).

A more detailed analysis of synaptic connections among vertically aligned neurons in *Pcdhg* cKO mice unveiled that the absence of γ-PCDH expression significantly increased synapse formation between cells separated vertically by 50–100 μm (24.0%, 50/208 in *Pcdhg* cKO mice vs. 9.6%, 20/208 in WT mice) (*Figure 2E*). Notably, this increased synapse formation was detected starting from P10 (16.1%, 32/199 in *Pcdhg* cKO mice *vs.* 8.7%, 20/230 in WT mice for P10–12; 21.6%, 22/102 in *Pcdhg* cKO mice vs. 12.8%, 29/227 in WT mice for P13–15). These time points correspond to the period when chemical synapses between cortical neurons become detectable, and this trend persisted throughout our measurements, spanning up to P20 (*Figure 2F*). Hence, our findings suggest that γ-PCDHs may play a role in preventing synapse formation from the early stages of neural development.

## Overexpression of γ-PCDHs decreases local synaptic connectivity in the mouse neocortex

To further determine the influence of γ-PCDHs on synaptic connectivity, we overexpressed randomly selected single or multiple γ-PCDH isoforms tagged with fluorescent proteins through in utero electroporation in mice. Subsequently, we performed a series of multiple whole-cell patch-clamp recordings to unveil how γ-PCDHs affect synaptic connectivity between neurons situated in layer 2/3 of the neocortex.

Firstly, to validate the overexpression effect, we conducted a battery of assays, including real-time quantitative reverse transcription PCR (qRT-PCR) (*Figure 3A and B*), single-cell RT-PCR (*Figure 3C and D*), and immunohistochemistry (*Figure 3—figure supplement 1A–C*). The qRT-PCR assays unequivocally confirmed that the electroporated isoforms, as opposed to the non-overexpressed ones from the contralateral side, exhibited significantly higher expression levels (*Figure 3A and B*). To assess how many isoforms were typically expressed in a given neuron when multiple plasmids were electroporated, we strategically tagged the first five isoforms with mNeongreen and the sixth with mCherry (*Figure 3—figure supplement 1A*). Employing a probability analysis based on the occurrence of yellow and red-only cells within the total electroporated neuron population (*Figure 3—figure supplement 1B*), we ascertained that each positive neuron expressed an average of 5.6 types of isoforms (*Figure 3—figure supplement 1C* and the top panel of *Figure 3—figure supplement 1D*). This result harmonized with data obtained from single-cell RT-PCR analysis, wherein an average of 5.3 types out

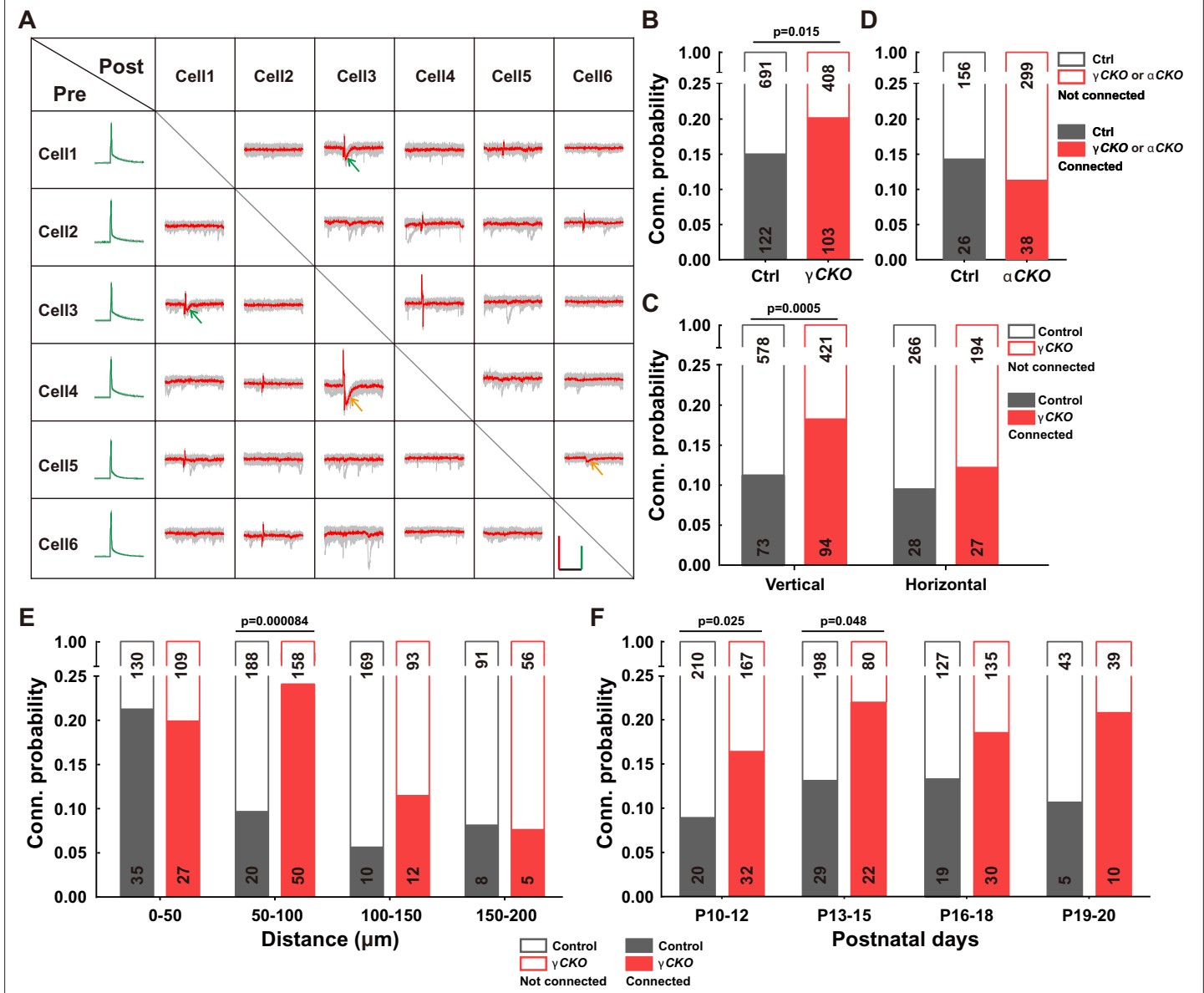

**Figure 2.** Increased synaptic connectivity in the absence of γ-PCDH. (**A**) Representative traces (red/green) from multiple electrode whole-cell patch-clamp recordings conducted on six neurons in layer 2/3 of the barrel cortex. Average traces are shown in red and green, with 10 original traces in gray. Positive evoked postsynaptic responses are indicated by arrows. Orange and green arrows denote unidirectional and bidirectional synaptic connections, respectively. Scale bars: 100 mV (green), 50 pA (red), and 50 ms (black). (**B, D**) Connectivity probability among nearby pyramidal cells in layer 2/3 of the barrel cortex in *Pcdhg* conditional knockout (cKO) (**B**), *Pcdha* cKO mice (**D**), and their littermate WT controls. γCKO: *Pcdhg* $^{fcon3/fcon3}$:: *Neurod6-cre* mice; αCKO: *Pcdha*$^{flox/flox}$:: *Neurod6-cre* mice; Ctrl: *Pcdhg*$^{+/+}$:: *Neurod6-cre* or *Pcdha*$^{+/+}$:: *Neurod6-cre* mice. (**C**) Connectivity probability among vertically or horizontally aligned neurons in layer 2/3 of the barrel cortex in *Pcdhg* cKO mice and their littermate WT controls. (**E**) Connectivity probability among vertically aligned pyramidal cells as a function of the distance between recorded pairs in *Pcdhg* cKO mice and WT mice. (**F**) Developmental profiling of connectivity probability among vertically aligned neurons in *Pcdhg* cKO mice and WT mice. Chi-square tests were used in (**B–F**) to calculate statistical differences.

The online version of this article includes the following source data and figure supplement(s) for figure 2:

**Source data 1.** Electrophysiological recording data for measuring synaptic connectivites between neurons in *Figure 2*.

**Figure supplement 1.** Multi-electrode whole-cell patch-clamp recording.

**Figure supplement 2.** Generation and characterization of *Pcdha*$^{flox/flox}$ mice.

**Figure supplement 2—source data 1.** Southern Blot gel images for *pcdha* knock in mice and expression level of *pcdha* from different genotypes of mice.

*Figure 2 continued on next page*

*Figure 2 continued*

**Figure supplement 3.** Top 10 off-target sequences of sgRNA_1.

**Figure supplement 3—source data 1.** Original sequencing data for top 10 off-target sequences of sgRNA_1 in *Figure 2—figure supplement 3* and sgRNA_2 in *Figure 2—figure supplement 4*.

**Figure supplement 4.** Top 10 off-target sequences of sgRNA_2.

of the six electroporated isoforms was detected from 19 neurons (*Figure 3C and D* and the bottom panel of *Figure 3—figure supplement 1D*). These single-cell RT-PCR findings further substantiate that the electroporation-introduced isoforms predominate within these neurons (*Figure 3D*).

Since the distribution of neocortical neurons across different layers significantly influences their synaptic connections (*He et al., 2015*), we meticulously examined the positions of these neurons relative to the pial surface after overexpression. Remarkably, we observed that overexpressing γ-PCDH isoforms did not induce any alterations in cell positioning within the neocortex compared to the control plasmids (*Figure 3—figure supplement 1E and F*).

Subsequently, through multiple recordings, we embarked on a quest to assess the impact of γ-PCDHs on synapse formation. Intriguingly, overexpressing one or six γ-PCDH variable isoforms in neurons significantly diminished the rate of synaptic connections among them (10.3%, 15/146 in expressing control plasmids; 1.9%, 4/216 in overexpressing one isoform; and 4.4%, 8/181 in overexpressing six isoforms, red bars in *Figure 3E*). However, when overexpressing γ-PCDH C4, we did not observe any significant effect on the synaptic connection rate (11.3%, 12/106 in neurons overexpressing γ-PCDH C4, *Figure 3E*).

To further exclude the potential influence of C-type γ-PCDHs in synapse formation, we employed a similar strategy in *Pcdhg* cKO mice, electroporating six variable isoforms. Remarkably, this overexpression also led to a reduction in the connection rate in *Pcdhg* cKO mice (6.1%, 12/198 in overexpressing six isoforms vs. 16.5%, 31/188 in expressing control plasmids), mirroring the outcome observed in WT littermate mice (6.6%, 8/121 in overexpressing six isoforms vs. 16.5%, 18/109 in expressing control plasmids) (*Figure 3F*). These compelling observations collectively underscore that overexpressing the variable isoforms, as opposed to the C-type isoform C4, leads to a decrease in synaptic connectivity within the mouse neocortex.

## Combinatorial expression of γ-PCDHs regulates synaptic connectivity in the mouse neocortex

Our quest to understand the influence of combinatorial expression patterns of γ-PCDH isoforms on synapse formation led us to conduct a sequential in utero electroporation at E14.5 and E15.5 (*Figure 4A*). In this intricate procedure, our objective was to deliberately manipulate the degree of similarity in expression between two distinct groups of neurons. To achieve this, we randomly selected isoforms, although with the notable exception of isoforms A5 and B8. These two isoforms were singled out due to their low expression levels as indicated by the single-cell sequencing data (*Figure 1D*, *Figure 1—figure supplement 1E*). The various isoforms were thoughtfully combined to create similarity levels spanning from 0% (indicating no overlap, complete dissimilarity between the two groups) to 100% (representing complete overlap, indicating that the two groups are entirely identical) (*Figure 4B*).

For the purpose of electroporation at E14.5 and E15.5, we employed fluorescent proteins mNeongreen and mRuby3 (*Bajar et al., 2016*) to tag isoforms, which allowed us to distinguish the cells electroporated on these respective days (*Figure 4A*). Subsequently, whole-cell patch-clamp recordings were performed on layer 2/3 neurons in the neocortex using acute brain slices containing both green and red cells from P10–14 pups. Each set of recordings encompassed at least one mNeongreen[+], one mRuby3[+], and one nearby control neuron without fluorescence (*Figure 4C*). The results were very clear: neurons sharing the same color displayed significantly lower connectivity (*Figure 4—figure supplement 1A*), aligning with our previous findings from single electroporation (as shown in *Figure 3E*). In the group characterized by a 100% similarity level, the connectivity rate between neurons with different colors that were electroporated on different days was also significantly lower compared to the control (3.7%, 5/134 in the complete-overlap group; 12.5%, 19/151 in control pairs, 100% in *Figure 4D*). However, as the similarity levels descended from 100% to 0%, the connectivity

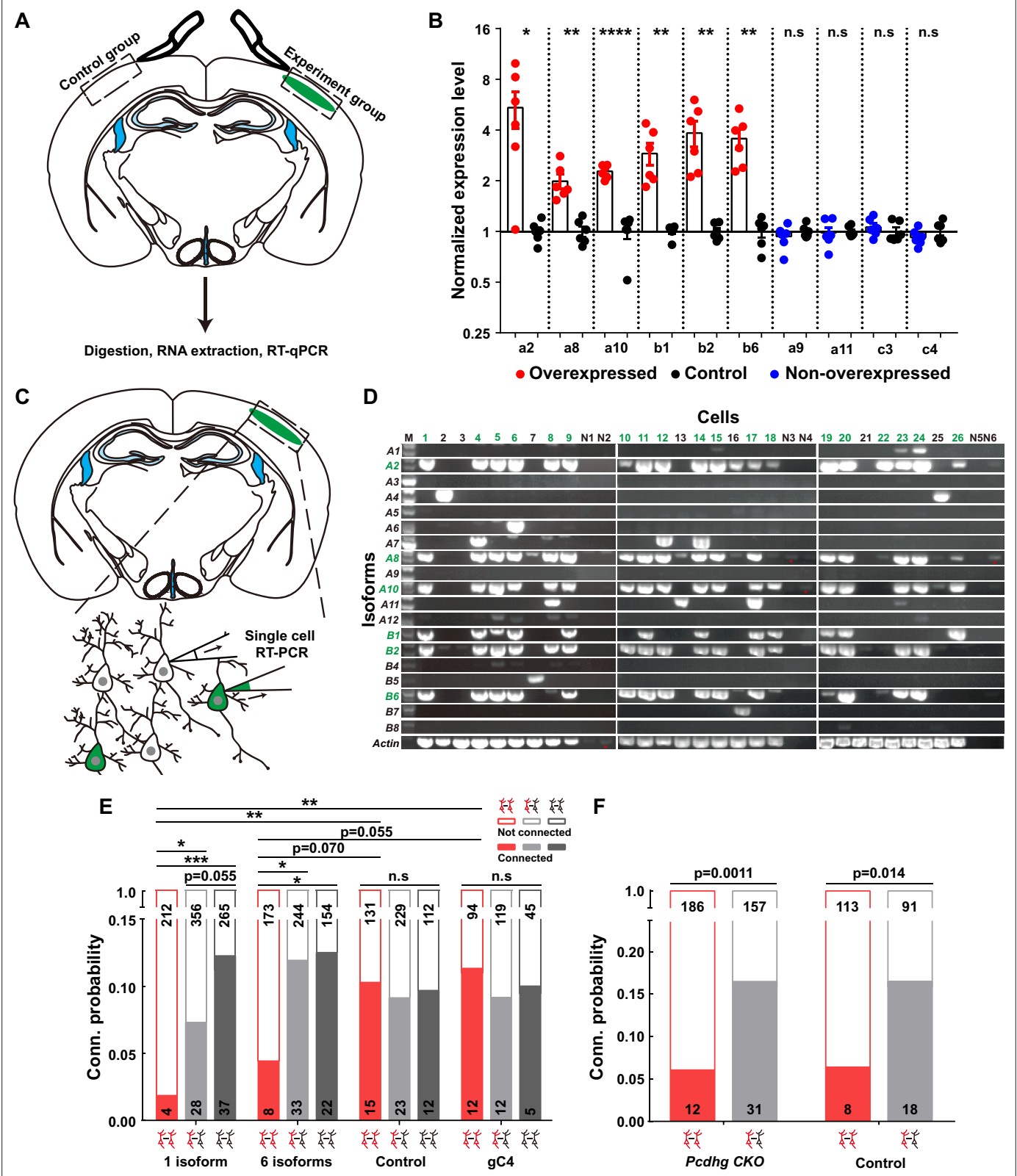

**Figure 3.** Overexpressing identical variables, but not C4, γ-PCDHs in neurons decreased their synaptic connectivity. (**A**) Schematic illustrating the brain regions selected for quantitative reverse transcription PCR (qRT-PCR) in both experimental and control groups. (**B**) qRT-PCR results showing overexpression levels in electroporated regions. Electroporated isoforms are indicated in red, control isoforms in blue, and contralateral sides used as controls are in black. Statistical analysis was conducted using Student's t-test, where *p<0.05, **p<0.01, ***p<0.001, and ****p<0.0001. (**C**) Diagram

*Figure 3 continued*

illustrating the process of cell extraction for single-cell RT-PCR assays. (**D**) Results of single-cell RT-PCR for γ-PCDH isoforms after electroporation. Neurons with fluorescence are highlighted in green, while nearby neurons without fluorescence are in black. Negative controls are labeled as N1–N6. Electroporated isoforms are shown in green, with red stars indicating faint signals in negative controls. (**E**) Impact of overexpressing one or six γ-PCDH isoforms on synaptic connectivity in WT mice. '1 isoform' represents *Pcdhga2*, and '6 isoforms' denote *Pcdhga2*, *Pcdhga8*, *Pcdhga10*, *Pcdhgb1*, *Pcdhgb2*, and *Pcdhgb6*. 'gC4' stands for *PcdhC4*, and 'Control' indicates plasmid vector without *Pcdhg* insertion. (**F**) The influence of overexpressing six γ-PCDH isoforms on synaptic connectivity in *Pcdhg* conditional knockout (cKO) mice. The same six isoforms used as in (**E**) were employed. *Pcdhg cKO: Pcdhg* conditional knockout mice; Control: WT littermates. Statistical differences between groups in (**E**) and (**F**) were determined using the chi-square test and false discovery rate (FDR, Benjamini–Hochberg method) correction.

The online version of this article includes the following source data and figure supplement(s) for figure 3:

**Source data 1.** RT-qPCR for testing overexpression level of different *pcdhg* isoforms in *Figure 3B*.

**Source data 2.** Gel images for the expressions of different *pcdhg* isoforms in *Figure 3D*.

**Source data 3.** Electrophysiological recording data for measuring synaptic connectivites between neurons in *Figure 3*.

**Figure supplement 1.** Characterizing the overexpression effect after electroporation.

**Figure supplement 1—source data 1.** Raw images and simulation code for *Figure 3—figure supplement 1*.

probabilities progressively reverted to the control level. The likelihood rebounded to 8.4% (14/165) for the pairs with a 33% similarity level and 10.3% (12/117) and 10.5% (12/114) for pairs with 11% or 0% similarity level, respectively (*Figure 4D*). This compelling observation illuminated a fundamental principle: it is the similarity level of γ-PCDH isoforms shared between neurons, rather than the absolute expression of the protein within individual neurons, that dictates the regulation of synaptic formation.

Remarkably, in line with the single-electroporation experiment (gray bars in *Figure 3E*), no significant changes were observed in the synaptic connectivity between electroporated neurons and nearby control neurons (*Figure 4—figure supplement 1B*). In summation, our findings illuminate a discernible negative correlation between the probability of synaptic connections and the similarity level of γ-PCDH isoforms expressed in neuron pairs (*Figure 4E*). These discoveries underline the significance of the diversified combinatorial expression of γ-PCDH isoforms in regulating synapse formation between adjacent pyramidal cells. Simply put, the more similar the patterns of γ-PCDH isoforms expressed in neurons, the lower the probability of synapse formation between them (*Figure 4F*).

## Discussion

Homophilic proteins cPCDHs are strong candidates for promoting synaptic specificity due to their combinatorial and stochastic expression pattern (*Yagi, 2012*; *Kohmura et al., 1998*; *Toyoda et al., 2014*). Our 5' end single-cell sequencing data provided solid evidence for the combinatorial expression pattern of γ-PCDH isoforms in neocortical neurons. We further demonstrated the critical role of this diversity in synaptic connectivity through three lines of evidence. Firstly, the absence of γ-PCDH significantly increased functional connectivity between adjacent neocortical neurons. Secondly, electroporation-induced overexpression of identical γ-PCDH variable isoforms in developing neurons markedly decreased their connectivity. Lastly, using sequential in utero electroporation with different batches of isoforms, we found that increasing the similarity level of γ-PCDH variable isoforms expressed in neurons led to a reduction in their synaptic connectivity. These findings suggest that γ-PCDHs regulate the specificity of synapse formation by preventing synapse formation with specific cells, rather than by selectively choosing particular targets. It remains to be studied whether the diversified patterns of γ-PCDH isoforms expressed in different neurons have additional coding functions for neurons beyond their homophilic interaction.

Stochastic and combinatorial expression patterns of cPCDH have been identified in Purkinje cells (*Esumi et al., 2005*; *Toyoda et al., 2014*) and olfactory sensory neurons (*Mountoufaris et al., 2017*). However, it is noteworthy that in serotonergic neurons only one isoform, Pcdhac2, has been mainly detected (*Chen et al., 2017*). In our study, utilizing 5' end single-cell sequencing, we have unveiled the stochastic and combinatorial expression patterns of variable γ-PCDH isoforms in neocortical neurons. These diverse observations across different cell types suggest that cPCDH diversity and the presence of ubiquitous C-type expression are not universal features throughout the brain (*Kiefer et al., 2023*). These distinct expression patterns of cPCDHs imply that this gene cluster might exert different roles

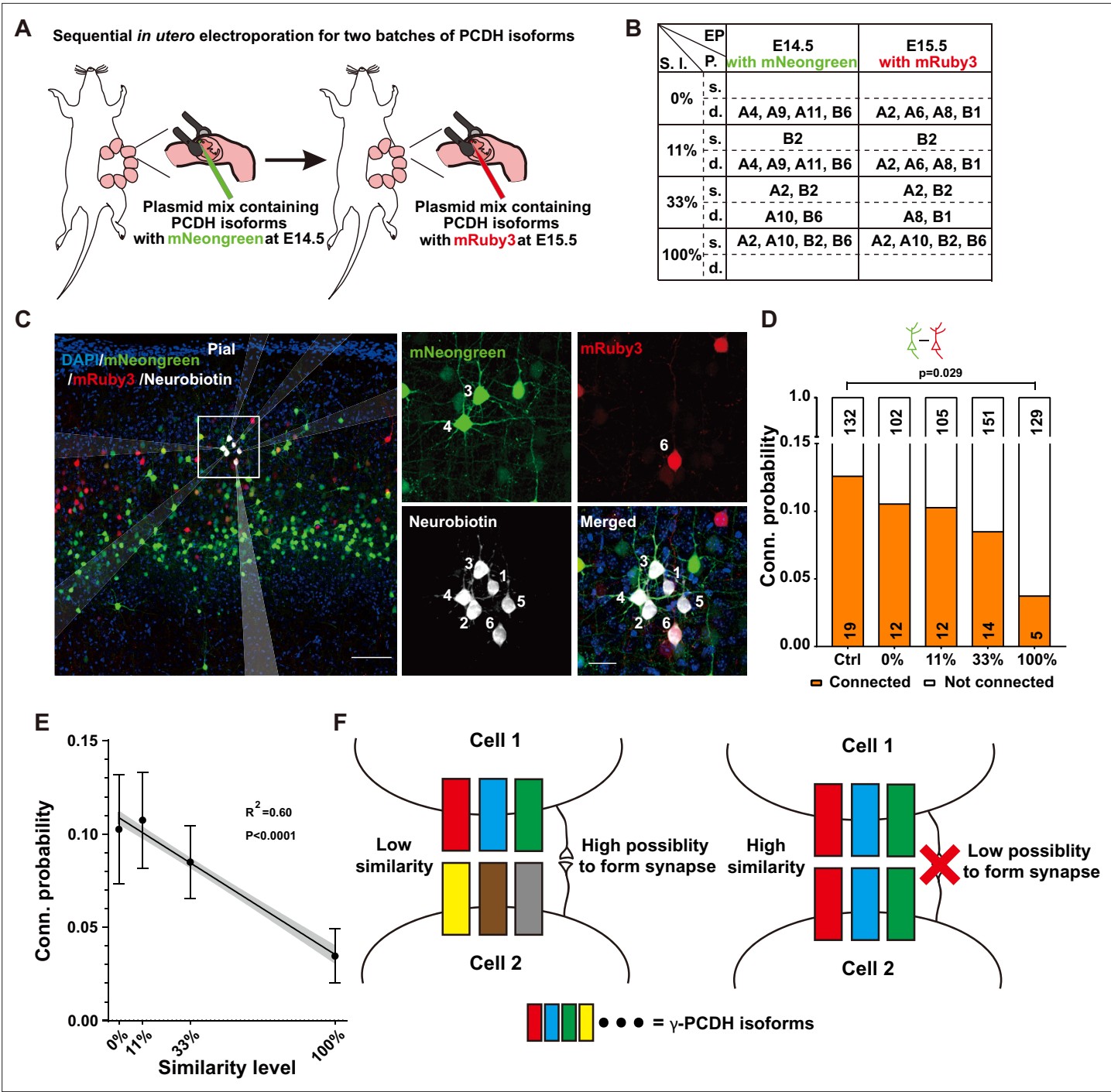

**Figure 4.** Diversified γ-PCDHs are critical for synapse formation in cortical neurons. (**A**) Diagram illustrating the sequential in utero electroporation process at E14.5 and E15.5. (**B**) Overview of the overexpressed γ-PCDH isoforms in different experiments, resulting in varying similarity levels between neurons. S.l.: similarity level; EP: electroporation; P.: plasmids mixture; s./d., same or different isoforms in two electroporations. (**C**) Sample image of recorded neurons after two rounds of electroporation at E14.5 (mNeongreen) and E15.5 (mRuby3). Neurons labeled as positive for mNeongreen (cells 3 and 4), mRuby3 (cell 6), and negative without fluorescence (cells 1, 2, and 5). Neurobiotin was used in the internal solution to label recorded neurons. The translucent arrows show the positions of the electrodes. Scale bar, left: 100 μm, right: 25 μm. (**D**) Connectivity probability for neuron pairs overexpressing different sets of γ-PCDH isoforms (labeled with different fluorescence) following sequential in utero electroporation. Statistical differences were determined using the chi-square test with false discovery rate (FDR) (Benjamini–Hochberg method) correction. (**E**) Correlation between the similarity level of overexpressed γ-PCDH combinations and the probability of synaptic connections. Each data point corresponds to the outcome of 100 bootstrapped samples derived from the source data presented in panel (**D**). Error bars indicate the standard deviation (SD) for each data point. The gray shaded area represents the 95% confidence interval of the curve fitting. (**F**) Graph summarizing the effect of γ-PCDHs on synapse formation.

*Figure 4 continued on next page*

*Figure 4 continued*

The online version of this article includes the following source data and figure supplement(s) for figure 4:

**Source data 1.** Electrophysiological recording data for measuring synaptic connectivites between neurons in *Figure 4D* and *Figure 4—figure supplement 1*.

**Figure supplement 1.** The effect of overexpressing γ-PCDHs on synaptic connectivity.

in shaping neural connections in various brain regions. Furthermore, compared to the SMART seq results from the Allen Institute's database (*Tasic et al., 2018*) and others (*Lv et al., 2022*) focusing on γ-PCDH, 5' end single-cell sequencing used in our study not only detected more isoforms in individual cells but also revealed more neurons expressing C-type isoforms. The application of this approach may offer valuable insights for studying the functions of cPCDHs in a broader neurological context.

Previous studies by Molumby et al. demonstrated that neurons from the neocortex of *Pcdhg* knockout mice exhibited significantly more dendritic spines, while neurons overexpressing a single γ-PCDH isoform had fewer dendritic spines in *Molumby et al., 2017*. Our recordings are consistent with these previous morphology studies. Tarusawa et al. revealed that the absence of the whole cluster of cPCDH affected synaptic connections among lineage-related cells (*Tarusawa et al., 2016*). More recently, overexpression of the C-type γ-PCDH isoform C3 also showed a negative effect on synapse formation within a defined clone (*Lv et al., 2022*). In our study, we further demonstrated an increased synaptic connection rate between adjacent pyramidal neurons in the neocortex of *Pcdhg* knockout mice, while it decreased between neurons overexpressing single or multiple identical γ-PCDH variable isoforms. These effects were not just limited to lineage-dependent cells. Together with previous findings (*Molumby et al., 2017*; *Tarusawa et al., 2016*), our observations solidify the negative effect of γ-PCDHs on synapse formation among neocortical neurons. Some subtle differences exist between our findings and previous recordings (*Tarusawa et al., 2016*; *Lv et al., 2022*). Tarusawa et al. demonstrated that the connection probability between excitatory neurons lacking the entire cPCDH cluster in layer 4 was approximately twofold higher at early stage P9–11, significantly lower at P13–16, and similar to control cells at P18–20 compared to *Tarusawa et al., 2016*. In our study, *Pcdhg*[-/-] pyramidal neuronal pairs consistently exhibited a higher connection probability from P10 to P20. Two potential reasons could explain these differences. Firstly, we only removed γ-PCDHs instead of the entire cPCDH cluster, which includes α, β, and γ isoforms. Secondly, γ-PCDH might have different functions in neurons located in layer 2/3 compared to layer 4. Lv et al. found that overexpression of C-type γ-PCDH C3 decreased the preferential connection between sister cells (*Lv et al., 2022*). However, our study demonstrated that only variable, but not C-type isoform C4, had a negative impact on synapse formation. This discrepancy might be attributed to the lineage relationship, which could have an unknown impact on synapse formation. Building upon previous findings, it is becoming increasingly evident that distinct C-type isoforms may play varied roles in shaping neural networks within the brain (*Garrett et al., 2019*; *Steffen et al., 2023*; *Meltzer et al., 2023*; *Lv et al., 2022*).

Since a single neuron can express multiple isoforms, deleting all γ-PCDH isoforms might mask the role of this combination. In this study, we manipulated the combinatorial expression patterns of γ-PCDH isoforms in nearby neocortical neurons through sequential in utero electroporation, expressing different batches of isoforms with adjustable similarities. We observed that when two neurons expressed identical variable isoforms (100% group), the likelihood of synapse formation between them was the lowest. As the similarity level between two cells decreased, with fewer shared isoforms, the connectivity probability increased. The connectivity probability between neurons with different variable isoforms (0% group) did not differ from the control pairs (without overexpression). However, the connections between overexpressed and control neurons were not affected under both 100 and 0% similarity conditions. These observations suggest that the similarity level, rather than the absolute expression of the protein, affects synapse formation between neurons. While we observed a negative correlation between expression similarity and the probability of connectivity among neocortical neurons, further investigation is needed to determine the precise cellular mechanisms underpinning this correlation. It is essential to explore whether this correlation arises directly from the formation of synapses or is a secondary effect resulting from cell positioning (*Lv et al., 2022*), synaptic pruning (*Kostadinov and Sanes, 2015*), or the influence of γ-PCDHs on the growth of axon/dendrite (*Molumby et al., 2017*; *Molumby et al., 2016*). Previous studies have established that the interplay between γ-PCDHs and neuroligin-1 plays a crucial role in the negative regulation of dendritic

spine morphogenesis (*Molumby et al., 2017*). Consequently, exploring whether the similarity in the expression of γ-PCDHs between two neurons influences their interaction with neuroligin-1 could yield valuable insights.

Our findings also demonstrated that the overexpression of multiple γ-PCDH variable isoforms in one neuron only affected its connection if the other neuron overexpressed an identical combination of γ-PCDH isoforms. This highlights the pivotal role of the diversified combinatorial expression of γ-PCDHs of neurons in selecting their synaptic partners in the mouse neocortex. Notably, while the overexpression of the γ-PCDH C4 isoform had no discernible effect on synaptic connectivity, over-expressing six variable isoforms resulted in a reduced connection rate in *Pcdhg* cKO mice. These observations underscore the critical role of variable isoforms, as opposed to the C-type isoform C4, in synapse formation within the mouse neocortex. Although the overexpression of the γ-PCDH C3 isoform has been shown to have a negative effect on synapse formation between sister cells, but no effect on synapses among non-clone cells in the neocortex (*Lv et al., 2022*), the distinct functions of individual C-type isoforms require further thorough examination. It is worth noting that our observations primarily stem from overexpression assays, providing insights into the effects of γ-PCDHs on synaptic connectivity. Exploring their impact under more physiological conditions using alternative approaches holds significant promise.

Furthermore, while the absence of γ-PCDHs causes significantly more synaptic formation among neocortical pyramidal neurons, evidence supports that their absence also leads to a significant reduction of synapse formation in other brain regions. For example, mice lacking γ-PCDHs exhibit fewer synapses in spinal cord interneurons (*Weiner et al., 2005*) and diminished astrocyte–neuron contacts in co-cultures from the developing spinal cord (*Garrett and Weiner, 2009*). Knocking down γ-PCDHs causes a decline in dendritic spines in cultured hippocampal neurons (*Suo et al., 2012*). The absence of γ-PCDHs leads to reduced dendritic arborization and dendritic spines in olfactory granule cells (*Ledderose et al., 2013*). Additionally, immuno-positive signals for γ-PCDHs are more frequently detected in mushroom spines than in thin spines (*LaMassa et al., 2021*). Moreover, our observations revealed that the absence of γ-PCDHs had a more pronounced impact on vertically aligned neurons than on horizontally situated pairs in the neocortex. These findings suggest that different mechanisms may be employed by synapses in different brain regions to achieve their specificity. Notably, different mechanisms have already been proposed for targeting specific inhibitory neural circuits in the neocortex, including 'on-target' synapse formation for targeting apical dendrites and 'off-target' synapse selective removal for somatic innervations (*Gour et al., 2021*).

In summary, our data demonstrate that the similarity level of γ-PCDH isoforms between neocortical neurons is critical for their synapse formation. Neurons expressing more similar γ-PCDH isoform patterns exhibit a lower probability of forming synapses with one another. This suggests that the presence of γ-PCDHs enables neocortical neurons to choose which neurons to avoid synapsing with, rather than selecting specific neurons to form synapses with. Whether there are specific attractive forces between cells to promote synaptic specificity remains an open question.

## Materials and methods

**Key resources table**

| Reagent type (species) or resource | Designation | Source or reference | Identifiers | Additional information |
|---|---|---|---|---|
| Strain, strain background (C57BL/6 *Mus musculus*) | *Pcdhg*<sup>fcon3/fcon3</sup> | CAS Center for Excellence in Brain Science and Intelligence Technology | | From Dr.Yifeng Zhang Jax: 012644; DOI:10.1242/dev.027912 |
| Strain, strain background (C57BL/6 *M. musculus*) | *Neurod6-cre (also known as Nex-cre)* | CAS Center for Excellence in Brain Science and Intelligence Technology | | From Dr.Zilong Qiu; DOI:10.1002/dvg.20256 |
| Strain, strain background (C57BL/6 *M. musculus*) | *Pcdha*<sup>flox/flox</sup> | Biocytogen Pharmaceuticals (Beijing) (*Figure 2D* and *Figure 2—figure supplements 2–4*) | | Established in this study |

*Continued on next page*

*Continued*

| Reagent type (species) or resource | Designation | Source or reference | Identifiers | Additional information |
|---|---|---|---|---|
| Strain, strain background (C57BL/6N *M. musculus*) | C57 | Vital River Laboratories | 213 | |
| Strain, strain background (CD-1(ICR) *M. musculus*) | ICR WT | Vital River Laboratories | 201 | |
| Antibody | Anti-RFP (rabbit polyclonal) | Rockland | 600-401-379 | IHC (1:1000) |
| Recombinant DNA reagent | CAG-pcdhga2-mNeongreen (plasmid) | This paper (*Figures 3B* and *4B*) | P80 | |
| Recombinant DNA reagent | CAG-pcdhga10-mNeongreen (plasmid) | This paper (*Figures 3B* and *4B*) | P81 | |
| Recombinant DNA reagent | CAG-pcdhgb2-mNeongreen (plasmid) | This paper (*Figures 3B* and *4B*) | P82 | |
| Recombinant DNA reagent | CAG-pcdhgb6-mNeongreen (plasmid) | This paper (*Figures 3B* and *4B*) | P83 | |
| Recombinant DNA reagent | CAG-pcdhga6-mNeongreen (plasmid) | This paper (*Figures 3B* and *4B*) | P115 | |
| Recombinant DNA reagent | CAG-pcdhga8-mNeongreen (plasmid) | This paper (*Figures 3B* and *4B*) | P116 | |
| Recombinant DNA reagent | CAG-pcdhgb1-mNeongreen (plasmid) | This paper (*Figures 3B* and *4B*) | P117 | |
| Recombinant DNA reagent | CAG-pcdhgC4-mNeongreen (plasmid) | This paper (*Figure 3E*) | P269 | |
| Recombinant DNA reagent | CAG-pcdhga2-mRuby3 (plasmid) | This paper (*Figure 4B*) | P84 | These plasmids made in this paper: The genes were cloned from WT C57BL/6N cDNA (whole brain) and inserted to the CAGGS plasmid backbone (AscI/NotI site), respectively |
| Recombinant DNA reagent | CAG-pcdhga10-mRuby3 (plasmid) | This paper (*Figure 4B*) | P85 | |
| Recombinant DNA reagent | CAG-pcdhgb2-mRuby3 (plasmid) | This paper (*Figure 4B*) | P86 | |
| Recombinant DNA reagent | CAG-pcdhgb6-mRuby3 (plasmid) | This paper (*Figure 4B*) | P87 | |
| Recombinant DNA reagent | CAG-pcdhga4-mRuby3 (plasmid) | This paper (*Figure 4B*) | P206 | |
| Recombinant DNA reagent | CAG-pcdhga7-mRuby3 (plasmid) | This paper (*Figure 4B*) | P207 | |
| Recombinant DNA reagent | CAG-pcdhga9-mRuby3 (plasmid) | This paper (*Figure 4B*) | P208 | |
| Recombinant DNA reagent | CAG-pcdhga11-mRuby3 (plasmid) | This paper (*Figure 4B*) | P209 | |
| Recombinant DNA reagent | CAG-pcdhga6-mCherry (plasmid) | This paper (*Figure 3—figure supplement 1A*) | P132 | |
| Commercial assay or kit | PrimeScript RT Master Mix | Takara | RR036A | |
| Commercial assay or kit | SYBR Green Real-time PCR Master Mix | Toyobo | QPK-201 | |
| Commercial assay or kit | Mouse T Cell Chromium Single Cell V(D)J Enrichment Kit | 10X Genomics | PN-1000071 | |
| Software, algorithm | MATLAB R2018b | MathWorks | | |

## Mice

All mice were handled in strict compliance with the approved protocol from the Animal Care and Use Committee of the Center for Excellence in Brain Science and Intelligence Technology/Institute of Neuroscience, Chinese Academy of Sciences (NA-022-2022), and the committee at Lingang Laboratory (NZXSP-2022-10). These mice were maintained in group housing conditions. $Pcdhg^{fcon3/fcon3}$, $Neurod6$-$cre$ mice are maintained as heterozygotes within the C57BL/6 background. F2 mice resulting from the cross of these two lines contained $Pcdhg^{fcon3/fcon3}$/$Neurod6$-$cre$ as experimental groups, and their WT littermates were used as controls for experiments. ICR mice from Vital River Laboratories were used in the overexpression experiments.

## Single-cell dissociation

At P11, an ICR mouse was perfused with dissociation-artificial cerebrospinal fluid (aCSF) that had been thoroughly oxygenated with a 95% $O_2$/5% $CO_2$ mixture prior to use. The dissociation-aCSF used was based on choline-modified aCSF (choline-aCSF), consisting of the following components (in mM): 120 choline chloride, 2.6 KCl, 26 $NaHCO_3$, 1.25 $NaH_2PO_4$, 15 D-glucose, 1.3 ascorbate acid, 0.5 $CaCl_2$, and 7 $MgCl_2$ (with an osmolarity range of 300–310 mOsm), with the addition of 50 µM DL-APV (Abcam/ ab120271) and 10 µM DNQX (Sigma/D0540).

The brain was promptly removed and immersed in dissociation-aCSF buffer at 0°C. Half of the neocortex was stripped off and cut into small, approximately 100-µm-diameter chunks. These chunks were then placed into a digestion buffer (dissociation-aCSF plus 100 units papain and 500 units DNase I) and allowed to incubate at 37°C for 60 min. Throughout the incubation, a gentle stream of 95% $O_2$/5% $CO_2$ was passed over the buffer. Following incubation, the sample was centrifuged at room temperature for 5 min at $300 \times g$, and the resulting precipitate was collected and resuspended in 2 mL of Buffer II (dissociation-aCSF with 0.05% BSA).

The sample was then carefully triturated in sequence using fire-polished Pasteur pipettes with diameters of 600 µm, 400 µm, and 200 µm until no large, visible chunks remained. To remove debris, a debris removal solution (130-109-398, Miltenyi Biotec) was used in accordance with the manufacturer's protocol. Subsequently, the sample was resuspended with 500 µL Buffer II, which included 2.5 µg DAPI, and filtered through a 40 µm cell strainer before FACS sorting. Living cells were sorted via FACS based on their 350 nm fluorescence, ensuring a living cell rate of over 85%. The sorted cells were collected into Buffer II and subsequently subjected to centrifugation at 0°C and $500 \times g$ to obtain a single-cell suspension (refer *Figure 1—figure supplement 1*).

## Library construction for 5' end single-cell sequencing

We adapted the Mouse T Cell Chromium Single Cell V(D)J Enrichment Kit (PN-1000071, 10X Genomics) to create a library for the *Pcdhg* gene cluster. Although the kit was originally designed for sequencing T-cell receptor (TCR) genes in immune cells, we customized it by using specific primers for *Pcdhg* to sequence the 5' ends of different isoforms within this gene cluster.

We initiated the process by generating nanoliter-scale gel beads-in-emulsion (GEMs) using the Chromium Next GEM Chip G. Within these GEMs, we conducted reverse transcription, template switching, barcoding, and transcript extension (*Figure 1—figure supplement 1B*). The GEMs were later disrupted to facilitate cDNA amplification (*Figure 1—figure supplement 1B*). Upon completing the amplification, we divided the sample into two parts. One part was reserved for constructing an expression library for other genes at their 5' ends, while the other was used for constructing the enriched *Pcdhg* library (*Figure 1—figure supplement 1C*).

For the construction of the *Pcdhg* library, we replaced the original Reverse Outer & Inner Primers in Mouse T Cell Mix 1 and 2 with the *Pcdhg* nested-PCR primer mix, designed to target the constant regions of *Pcdhg*. This mix comprised the outer primer R1, situated within exon 4, and the inner primer R1, spanning exons 2 and 3 of the pcdhg gene. The complete sequences of the primers used can be found in the primer list in *Supplementary file 1*.

Following the enrichment of *Pcdhg*, we adhered to the manufacturer's protocol for the remaining steps of library construction (*Figure 1—figure supplement 1C*, left). For the construction of the 5' gene expression library, the amplified cDNA was immediately incorporated into the subsequent procedure in accordance with the kit's manual (*Figure 1—figure supplement 1C*, right). The two

libraries were sequenced independently and subsequently intersected based on the cell barcodes (*Figure 1—figure supplement 1D*).

## Sequencing data profiling

We obtained 212.66 GB clean reads for the 5' gene expression library and 183.30 GB for the 5' *pcdhg* expression library using the NovaSeq 6000 sequencing platform (Illumina, Novogene). After alignment with Cellranger v3.0.2 (10X Genomics) from the 5' gene expression sequencing data, we identified 23,753 cells (*Figure 1—figure supplement 1D*). These cells had an average of 29,843 reads per cell, and sequencing saturation reached 51.1%.

Cells were selected based on the criteria that feature RNA was within the range of 7000–1800, and mitochondrial RNA was less than 7%, resulting in 20,419 cells (*Figure 1—figure supplement 2A*). Subsequently, 2981 potential doublets (15%) were removed using Doubletfinder, and the remaining cells were clustered using Seurat2 (resolution = 2.4, optimized by the Clustree algorithm, *Figure 1—figure supplements 1D and 2B*). After clustering, selected genes, including *Slc17a7*, *Neurod2*, *Neurod6*, *Slc32a1*, *Gad1*, and *Satb2* for neurons, and *Sox2*, *Car2*, *Ascl1*, *Aldh1l1*, *Pdgfra*, *Pdgfrb*, *Olig1*, *C1qa*, and *Dlx2* for non-neurons, were used to determine the identity of each cluster (*Figure 1—figure supplements 2–4*). Upon separating non-neurons from neurons, neurons were further divided into 15 clusters (clusters 0–14) (resolution = 0.4, optimized by the Clustree algorithm, *Figure 1—figure supplement 4A*). Genes *Gad1*, *Gad2*, *Slc32a1*, *Slc17a1*, *Lamp5*, *Ndnf*, *Sncg*, *Vip*, *Sst*, *Pvalb*, *Cux2*, *Rorb*, *Fezf2*, *Sulf1*, *Foxp2*, *Mbp*, *Cldn5*, *Ctss*, and *C1qa* were used to identify the property of these clusters (*Figure 1—figure supplement 4B*). Cluster 12 was classified as atypical neuron due to its high expression of *C1qa* and was excluded from further analysis.

To analyze the 5' *Pcdhg* expression sequencing data, we used Bowtie with the reference sequence of *Pcdhg* variable axons from all isoforms. Cell barcodes and UMI sequences were extracted from the aligned reads using a custom script in MATLAB (R2018b, MathWorks). The *Pcdhg* expression matrix was generated from the intersection of data from the 5' gene expression library and the *Pcdhg* enrichment library (*Figure 1—figure supplement 1D*). In the output matrix, the *Pcdhg* expression level was represented by the number of UMIs. The *Pcdhg* expression matrix was filtered with a UMI count greater than 1 and binarized for subsequent analysis (*Figure 1—figure supplement 1D*). Various UMI thresholds for the total *Pcdhg* were applied to assess cell fractions for all isoforms in individual cells (*Figure 1D*, *Figure 1—figure supplement 1E*). A UMI count of greater than 10 was chosen for the similarity level analysis in 2834 neurons to ensure the quality of the assay.

## Definition of the similarity level

The similarity level was computed using the formula $\frac{A \cap B}{A \cup B}$, where $A$ and $B$ represent the numbers of γ-PCDH variable isoforms (excluding C-type isoforms C3, C4, and C5) expressed in two compared cells; $A \cap B$ represents the number of isoforms shared in both cells; and $A \cup B$ equals to the total number of isoforms expressed in these two cells.

In *Figure 1—figure supplement 5*, we also used the Euclidean distance and the normalized Euclidean distance to measure the similarity between two neurons. The Euclidean distance is a widely used method for describing the spatial distance between two nodes in a multidimensional space. In our dataset, each neuron represents one node in a 19-dimensional space, with each axis representing one variable isoform. The coordinate of each node in each axis is determined by the expression level of the corresponding isoform (counted by UMI). If a cell pair has a lower Euclidean distance, then these two cells express a more similar combination of γ-PCDH isoforms.

The Euclidean distance is calculated as follows:

$$\text{Euclidean distance} = \sqrt{\sum \left( \Delta^2_{UMIcounts(pcdhga1)} + \Delta^2_{UMIcounts(pcdhga2)} + \cdots + \Delta^2_{UMIcounts(pcdhgb8)} \right)}$$

However, if two cells have a substantial difference in total expression levels, the Euclidean distance may not be sufficiently sensitive. Therefore, we also explored the 'normalized Euclidean distance' to better assess the dissimilarity between two neurons. After normalizing the total expression level of variable isoforms (19 in total) in each cell to 1, we calculated the normalized Euclidean distance. The normalized Euclidean distance falls within the range of [0, √2].

## Co-occurrence

We adopted a concept from Wada's study to investigate the potential interactions among different isoform expressions at the population level. Co-occurrence analysis was used to determine whether each isoform expresses independently.

To perform this analysis, we first generated an expected matrix by shuffling data 100 times, drawing from different clusters (or the total population), assuming that all the isoforms were expressed randomly based on their expression frequencies. Subsequently, we plotted the fraction of cells with various numbers of isoform types from both the observed expression matrix (observed data) and the expected matrix (shuffled data) for the entire population (*Figure 1F*, *Figure 1—figure supplement 7*).

We then computed the variance of this distribution for cells in different clusters or the total population. The expected variance was determined using the following formula, taking into account the expression probability of each isoform (*pi*):

$$\sigma^2_{expected} = \sum_{i=1}^{N_{isoforms}} p_i \left(1 - p_i\right)$$

Subsequently, we used a Z-test to evaluate the statistical difference between the observed variance and the expected variance. If the observed variance was found to be higher than the expected variance, it indicated that the genes were co-occurring.

## Generation of *pcdhα* cKO mice

*Pcdha* cKO mice were generated using the CRISPR/Cas9 approach by Biocytogen Pharmaceuticals (Beijing). Candidate sgRNAs for introns 1 and 2 of *Pcdha1* were designed using the CRISPR design tool (http://crispr.mit.edu) and screened for on-target activity using the Universal CRISPR Activity Assay kit (UCATM, Biocytogen Pharmaceuticals [Beijing] Co., Ltd).

To minimize random integrations, a circular donor vector was employed. The gene-targeting template contained a 5' homologous arm, the target fragment (exon 2), and a 3' homologous arm. This template was used to repair the double-strand breaks generated by Cas9/sgRNA. Cas9 mRNA, the template, and sgRNAs were co-injected into the cytoplasm of one-cell stage fertilized C57BL/6 egg. These injected eggs were then implanted into the oviducts of Kunming pseudopregnant females to produce F0 mice.

F0 mice with the correct genotype were bred with C57BL/6 mice to establish germline-transmitted F1 heterozygous mice. The PCR-positive F1 mice were further confirmed with Southern blot analysis (*Figure 2—figure supplement 2*). DNA sequencing was used to confirm the absence of off-target effects (*Figure 2—figure supplements 3 and 4*). Exon 2 (first constant exon) of *Pcdha genes* will be removed with the assistance of Cre, resulting in an 813 aa (796aa native, 17aa in-frame shifted) truncated protein that may be subject to nonsense-mediated decay. The deletion of first constant exon will affect all *Pcdha* isoforms.

## Southern blot

The digoxin (DIG)-labeled probes were synthesized through PCR using the PCR DIG Probe Synthesis kit (11636090910, Roche), with genomic DNA extracted from mouse tails and digested with BamHI or StuI (R0136, R0187, New England BioLabs). The DNA was separated on a 1% agarose gel and transferred to a positively charged nylon membrane (HyBond N+, Amersham). The hybridization process was carried out at 42°C overnight using the DIG Easy Hyb Granules (11796895001, Roche). Following hybridization, the signals were subsequently amplified using the DIG Luminescent Detection kit (11363514910, Roche).

## sgRNA off-target test

Potential sgRNA off-target sites were identified using the website https://crispr.cos.uni-heidelberg.de/. Subsequently, primers (available in the provided list in *Supplementary file 1*) were designed based on the website https://crispr.bme.gatech.edu/ to target the top 10 off-target regions, aiming for a product size of approximately 600 bp.

For the experimental process, male *pcdha⁺/flox* mice were bred with female *pcdha⁺/flox* mice. From the resulting offspring, F1 generation homozygotes, heterozygote littermates, and WT littermates

were each selected for genome DNA extraction using the Mouse direct PCR Kit (B40018, Bimake). Following DNA extraction, PCR amplification was carried out, and the resulting PCR products were subsequently subjected to sequencing (Sangon, Shanghai).

## Acute slice preparation

The brains of mice aged between P9 and P14 were extracted immediately after sacrifice and placed on ice. Coronal sections with a thickness of 300 μm were prepared using a vibratome (VT1200S, Leica). The sections were cut in choline-aCSF containing the following concentrations (in mM): 120 choline chloride, 2.6 KCl, 26 NaHCO$_3$, 1.25 NaH$_2$PO$_4$, 15 D-glucose, 1.3 ascorbate acid, 0.5 CaCl$_2$, and 7 MgCl$_2$. This solution maintained an osmolarity of 300–310 mOsm. The slices were then transferred to a chamber with normal aCSF containing (in mM) 126 NaCl, 3 KCl, 26 NaHCO$_3$, 1.2 NaH$_2$PO$_4$, 10 D-glucose, 2.4 CaCl$_2$, and 1.3 MgCl$_2$ (with an osmolarity of 300–310 mOsm) and incubated at 32°C for 30 min. Then, they were maintained at room temperature until recording.

For mice older than P14, pentobarbital sodium was administered for anesthesia via intraperitoneal injection (50–90 mg/kg). Following anesthesia, the brains were then perfused with 0°C NMDG-aCSF containing (in mM) 93 N-methyl-D-glucamine, 2.5 KCl, 1.25 NaH$_2$PO$_4$, 30 NaHCO$_3$, 25 D-glucose, 20 HEPES, 5 Na-ascorbate, 3 Na-pyruvate, 2 thiourea, 10 MgSO$_4$, 0.5 CaCl$_2$, and 12 N-acetyl-L-cysteine (with a pH of 7.3–7.4 and an osmolarity of 300–310 mOsm). Subsequently, the brains were coronally sectioned, following the same procedure as described above. The slices were incubated in NMDG-aCSF at 37°C for 12–15 min, then transferred to normal aCSF, and kept at room temperature for 1 hr before recording.

It is worth noting that all of the solutions mentioned were continuously oxygenated with a mixture of 95% O$_2$/5% CO$_2$.

## Electrophysiology

During recording, the slices were consistently perfused with normal aCSF maintained at 30°C and oxygenated with a mix of 95% O$_2$ and 5% CO$_2$. Cell visualization within the slices was performed using a differential interference contrast microscope (BX51W1, Olympus) equipped with both a ×5/NA 0.1 objective and a ×40/NA 0.8 W objective.

Glass pipettes, with resistance at 8–12 MΩ, were prepared with a Flaming/Brown micropipette puller (P-97, Sutter Instrument). The internal pipettes solution contains (in mM) 126 potassium gluconate, 2 KCl, 2 MgCl$_2$, 10 HEPES, 0.2 EGTA, 4 Na$_2$-ATP, 0.4 Na$_3$-GTP, 10 creatine phosphate, and 10 ng/mL Alexa Fluor 405 Cadaverine. Multiple whole-cell patch-clamp recordings, involving up to six channels, were conducted using amplifiers (Multiclamp 700B, Molecular Devices) and Digidata (Digidata 1550A, Molecular Devices). Subsequently, the data were analyzed using pClamp9 (Molecular Devices).

Electrophysiological data underwent low-pass Bessel filtering at 2 kHz and were digitized at 20 kHz. The feedback resistor was set at 5 GΩ. All the recordings had a series resistance less than 50 MΩ. Neuronal connectivity was assessed by injecting current to one of the recorded cells under the current-clamp mode to evoke action potentials. Simultaneously, evoked excitatory postsynaptic currents (EPSCs) in other receiver cells were examined under the voltage-clamp mode, maintaining a holding potential of –70 mV.

To determine the presence of meaningful EPSC, the size of currents in receiver cells was quantified 1 s before stimulation to obtain the average and standard deviation (SD). A current within three mini-second after stimulation that was three times SD greater than the average and exhibited minimal jittering among different trials (shorter than 0.2 ms) was considered a meaningful EPSC. Each neuron pair in every recording was tested at least 10 times in both directions.

## In utero electroporation

For pregnant ICR mice or pcdhg cKO mice, the date of plug detection was designated as embryonic day 0.5 (E0.5). At the gestational stages of E14.5 or E15.5, the pregnant mice were anesthetized using 5% isoflurane and subsequently maintained under 1.0% isoflurane with a gas flow rate of 0.8–1 L/min. Following a surgical incision, the uterine horns were exposed, and a plasmid mixture with a total

concentration of 5 mg/mL plasmid (equal molar mixture if there are multiple plasmids) was injected into the embryonic lateral ventricle using a glass pipette. This plasmid mixture included 1 mg/mL Fast green (Sigma). After the injection, the utero was clamped with electrodes with a diameter of 7 mm (45-0118, Tweezertrodes), and five 30 V pulses of 50 ms duration were delivered with 1 s intervals.

In sequential in utero electroporation experiment, surgeries were performed at E14.5 and E15.5, and the plasmids were injected into the same lateral ventricle and delivered at a similar angle. Following the procedure, lincomycin hydrochloride and lidocaine hydrochloride gel were applied to the wound after stitching it up.

## qRT-PCR

For the confirmation of *pcdhg* overexpression (*Figure 3A and B*), the brains of mice electroporated at P11–14 were promptly removed at sacrifice and placed into 4°C choline-aCSF. The specific target region was then identified using a fluorescent microscope, and approximately 500-µm-thick coronal slices were prepared. Tissue blocks containing the fluorescently labeled region, as well as the contra-lateral control side, were extracted from the same coronal slice and immediately transferred into an RNA extraction buffer.

For *Pcdha* knockout confirmation (*Figure 2—figure supplement 2D*), the semi-cortex of P11 mice was isolated and transferred into an RNA extraction buffer. Tissue RNA was extracted using the Total RNA extraction Kit (R4011-02, Magen) and then reverse transcribed using the PrimeScript RT Master Mix (RR036A, Takara).

Primers, the sequences of which can be found in the primers list (*Supplementary file 1*), were designed with the assistance of PrimerBlast (NCBI). SYBR Green Real-time PCR Master Mix (QPK-201, Toyobo) and the LightCycler 480 II (Roche) were employed for qPCR.

The expression level was calculated as $2^{Ct(Pcdhg)-Ct(GAPDH)}$, and the normalized expression level (*Figure 3B*) was determined by dividing the expression level of the target fluorescent region by the expression level of the contralateral control region.

## Estimating the number of overexpressed *pcdhg* isoforms

Five plasmids, each containing one of the following isoforms *pcdhga*2, *pcdhga8*, *pcdhga10*, *pcdhgb1*, and *pcdhgb2*, tagged with mNeongreen were equimolarly mixed. A sixth plasmid, pCAGGS-*pcdhga6*-p2A-mCherry, was included for electroporation at E15.5. The mice were sacrificed at P7, and their brains were sectioned coronally with a vibratome at a thickness of 60 µm. The mCherry signal was enhanced by using an anti-RFP antibody (Rockland 600-401-379) to achieve a brightness similar to mNeongreen. The images were captured using a confocal microscope (Nikon C2, ×20, NA 0.75, *Figure 3—figure supplement 1A*).

The co-transfection rate was calculated by the following formula. For simplicity, we will use two plasmids first as an example.

When two plasmids, one containing GFP and the other RFP, were used for an electroporation at a 1:1 molar ratio:

$$R_{GFP-only} = \left(\frac{1}{2}\right)^n \tag{1}$$

$$R_{RFP-only} = \left(\frac{1}{2}\right)^n \tag{2}$$

$$R_{co} = 1 - \left(\frac{1}{2}\right)^n - \left(\frac{1}{2}\right)^n = 1 - \left(\frac{1}{2}\right)^{n-1} \tag{3}$$

The percentages of cells that only express one fluorescent protein in all transfected cells (the percentage of green or red cells in all transfected cells) are represented by $R_{GFP-only}$ and $R_{RFP-only}$, respectively. The percentages of cells that co-express both fluorescent proteins (the percentage of yellow cells in all transfected cells) are represented by $R_{co}$. The variable $n$ in the formula is the number of received plasmid for each cell.

When six plasmids, five tagged with GFP and one with RFP, were electroporated with equal molar ratio:

$$R_{GFP-only} = \left(\frac{5}{6}\right)^n \tag{4}$$

$$R_{RFP-only} = \left(\frac{1}{6}\right)^n \tag{5}$$

$$R_{co} = 1 - \left(\frac{5}{6}\right)^n - \left(\frac{1}{6}\right)^n \tag{6}$$

$$R_{Red\ in\ total} = R_{co} + R_{RFP-only} = 1 - \left(\frac{5}{6}\right)^n \tag{7}$$

Note that formulas (*Equations 1–7*) only describe the most idealized situation where $n$ is a fixed number for every cell. However, when these formulas were applied to fit our data, they did not provide a good fit. We recognized that $n$ was not a fixed number for every cell, and we hypothesized that $n$ might obey a normal distribution ($n \sim N\left(\mu, \sigma^2\right)$). A normal distribution is characterized by two parameters: the mean ($\mu$) and the standard error ($\sigma$). Therefore, we modified functions (5) and (7) to incorporate the idea that $n$ follows a normal distribution.

$$R_{RFP-only} = \frac{\sum\limits_{n=1}^{k} 1 - \left(\frac{1}{6}\right)^n}{k}, n \sim N\left(\mu, \sigma^2\right)' \tag{8}$$

$$R_{Red\ in\ total} = \frac{\sum\limits_{n=1}^{k} 1 - \left(\frac{5}{6}\right)^n}{k}, n \sim N\left(\mu, \sigma^2\right)' \tag{9}$$

The parameter $k$ in the formulas represents the total number of affected cells. We simulated $n$ by systematically varying the mean ($\mu$) and standard error ($\sigma$) with integer values and found that the mean value of 18 ($\mu = 18$) and the standard error of 6 ($\sigma = 6$) closely matched the experimental data (*Figure 3—figure supplement 1C*). We then simulated 10,000 cells following this distribution and counted the number of expressed isoform types in each cell. Ultimately, we found that each affected neuron expressed an average of 5.6 isoform types (*Figure 3—figure supplement 1D*).

## Single-cell RT-PCR

For single-cell RT-PCR, we used the SuperScript II Kit (18064014, Invitrogen). Neurons were collected from acute brain slices using a glass pipette with a resistance of 3–4 MΩ, pre-filled with approximately 1 µL of aCSF. The cell contents were then gently expelled into a PCR tube containing 0.5 µL Rnaseout (10777019, Invitrogen), and the samples were promptly frozen with liquid nitrogen.

Subsequently, RT mix1 (~1 µL sample, 0.5 µL Rnaseout, 10 mM dNTP, 1 µL primer mix [GR1 and β-actin R1, 2.5 µM], 1.5 µL ddH$_2$O) was added to the sample tubes and kept at 65°C for 5 min. The tubes were then quickly chilled on ice. RT mix2 (1 µL 10× RT buffer, 2 µL 25 mM MgCl$_2$, 1 µL 0.1 M DTT, 0.5 µL Rnaseout, and 0.5 µL SuperScript II) was added to the sample/mix1 for the reverse transcription, which took place at 50°C for 60 min, followed by 85°C for 5 min.

For the nested PCR, we employed LATaq Kit (RR002B, TaKaRa). The primer mix for the first round of the PCR contained forward primers *Gb* F1a, *Gb* F1b, *Ga* F1a, *Ga* F1b, *Ga* F1c, *Ga* F1d, and reverse primer *GR*1. Another primer mix, consisting of a forward primer corresponding to each isoform and reverse primer *GR*2, was used for the second round of PCR. The sequences of each primer are listed in *Supplementary file 1*.

Due to the high sensitivity achieved after two rounds of PCR, faint bands appeared in some negative controls, leading to potential false-positive signals (indicated by the red stars in *Figure 3D*). A true-positive signal was defined as a band with at least five times higher intensity than the false-positive band.

## Bootstrap resampling

We employed bootstrap resampling using the 'bootstrp' function in MATLAB (R2018b, MathWorks), setting the 'nboot' parameter to 100.

## Acknowledgements

We thank Dr. Yifeng Zhang for providing the *Pcdhg*<sup>fcon3/fcon3</sup> mouse line and Dr. Zilong Qiu for *Neurod6-cre (*also known as *Nex-cre)* mouse line, Dr. Jun Chu for sharing plasmids with mNeongreen and mRuby3, and other members in Xu lab for their discussions and technique supports. We are grateful to Prof. Mu-ming Poo and Song-hai Shi for critical reading of the manuscript. This work was supported by grants from the Training Program of the Major Research Plan of the National Natural Science Foundation of China, grant no. 91632101; Strategic Priority Research Program of the Chinese Academy of Sciences, grant no. XDB32010100; National Natural Science Foundation of China project 31671113; Shanghai Municipal Science and Technology Major Project, grant no. 2018SHZDZX05, the State Key Laboratory of Neuroscience and the Lingang Laboratory, grant no. LG-GG-202201-01.

## Additional information

### Funding

| Funder | Grant reference number | Author |
|---|---|---|
| National Natural Science Foundation of China | 91632101 | Hua-tai Xu |
| National Natural Science Foundation of China | 31671113 | Hua-tai Xu |
| State Key Laboratory of Neuroscience and the Lingang Laboratory | LG-GG-202201-01 | Hua-tai Xu |
| Shanghai Municipal Science and Technology Major Project | 2018SHZDZX05 | Hua-tai Xu |

The funders had no role in study design, data collection and interpretation, or the decision to submit the work for publication.

### Author contributions
Yi-jun Zhu, Data curation, Formal analysis, Investigation, Methodology, Writing - original draft; Cai-yun Deng, Funding acquisition, Investigation; Liu Fan, Investigation, Methodology; Ya-Qian Wang, Methodology; Hui Zhou, Resources; Hua-tai Xu, Conceptualization, Resources, Data curation, Formal analysis, Supervision, Investigation, Methodology, Writing - original draft, Project administration, Writing - review and editing

### Author ORCIDs
Yi-jun Zhu (iD) http://orcid.org/0009-0002-8384-1878
Hua-tai Xu (iD) http://orcid.org/0000-0002-1113-0455

### Ethics
All mice were handled in strict compliance with the approved protocol from the Animal Care and Use Committee of the Center for Excellence in Brain Science and Intelligence Technology/ Institute of Neuroscience, Chinese Academy of Sciences (NA-022-2022), and the committee at Lingang Laboratory (NZXSP-2022-10).

Reviewer #1 (Public Review): https://doi.org/10.7554/eLife.89532.3.sa1
Reviewer #2 (Public Review): https://doi.org/10.7554/eLife.89532.3.sa2
Author Response https://doi.org/10.7554/eLife.89532.3.sa3

## Additional files

### Supplementary files
• Supplementary file 1. Primers list. (1) Primers for 'library construction for 5' end single-cell sequencing'. (2) Primers for confirming the generation of pcdha cKO mice. (3) Primers for qRT-PCR. (4) Primers for single-cell RT-PCR.

- MDAR checklist

## Data availability

Source code has been provided for Figrue 1. Source data has been provided for all main figures and Figure 2—figure supplements 2,3 and Figure 3—figure supplement 1. The 5' single-cell RNA seq data for PCDHgamma expression in neocortical neurons have been uploaded to Dryad at https://doi.org/10.5061/dryad.931zcrjm7.

The following dataset was generated:

| Author(s) | Year | Dataset title | Dataset URL | Database and Identifier |
|-----------|------|---------------|-------------|-------------------------|
| Xu HT | 2024 | Combinatorial expression of gamma-protocadherins regulates synaptic connectivity in the mouse neocortex | https://doi.org/10.5061/dryad.931zcrjm7 | Dryad Digital Repository, 10.5061/dryad.931zcrjm7 |

The following previously published dataset was used:

| Author(s) | Year | Dataset title | Dataset URL | Database and Identifier |
|-----------|------|---------------|-------------|-------------------------|
| Tasic B, Yao Z, Graybuck LT | 2018 | WHOLE CORTEX & HIPPOCAMPUS - SMART-SEQ (2019) WITH 10X-SMART-SEQ TAXONOMY (2021) | https://portal.brain-map.org/atlases-and-data/rnaseq/mouse-whole-cortex-and-hippocampus-smart-seq | Allen Brain Map: Cell Types Database, mouse-whole-cortex-and-hippocampus-smart-seq |

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
