## [Editor Report · eLife assessment]

The authors used an innovative modified 10X genomic sequencing method to detect cPCDHg is-forms in pyramidal neurons. With **solid** electrophysiological recordings, they showed that neurons expressing the same sets of cPCDHg isoforms are less likely to form synapses with each other. These **valuable** findings confirms previous results and extend our understanding of cPCDHg diversity and neuronal connectivity.

---

## [Referee Report · Reviewer #1 (Public Review)]

The manuscript by Zhu and colleagues aimed to clarify the importance of isoform diversity of PCDHg in establishing cortical synapse specificity. The authors optimized 5' single-cell sequencing to detect cPCDHg isoforms and showed that the pyramidal cells express distinct combinations of PCDHg isoforms. Then, the authors conducted patch-clamp recordings from cortical neurons whose PCDHg diversity was disrupted. In the elegant experiment in Figure 3, the authors demonstrated that the neurons expressing the same sets of cPCDHg isoforms are less likely to form synapses with each other, suggesting that identical cPCDHg isoforms may have a repulsive effect on synapse formation. Importantly, this phenomenon was dependent on the similarity of the isoforms present in neurons but not on the amount of proteins expressed.

The authors have addressed most criticisms raised in the initial review and the manuscript has improved significantly. One of the major concerns in the first review was whether PCDHg isoforms, which are expressed at a much lower level than C-type isoforms, have true physiological significance. The authors conducted additional experiments to address this point by using PCDHg cKO and provided convincing data supporting their conclusion. The results from PCDHg C4 overexpression, showing no impact on synaptic connectivity, further clarified the importance of isoforms. The limitation of the paper is that most experiments relied on overexpression of isoforms. Whether the isoform diversity is necessary for the synapse refinement in a physiological condition remains further clarification.

---

## [Referee Report · Reviewer #2 (Public Review)]

This short manuscript by Zhu et al. describes an investigation into the role of gamma protocadherins in synaptic connectivity in the mouse cerebral cortex. First, the authors conduct a single-cell RNA-seq survey of postnatal day 11 mouse cortical neurons, using an adapted 10X Genomics method to capture the 5' sequences that are necessary to identify individual gamma protocadherin isoforms (all 22 transcripts share the same three 3' "constant" exons, so standard 3'-biased methods can't distinguish them). This method adaptation is an advance for examining individual clustered protocadherin transcripts, and it is helpful to publish the method in this manuscript. The results largely confirm what was known from other approaches, which is that a few of the 19 A and B subtype gamma protocadherins are expressed in an apparently stochastic and combinatorial fashion in each cortical neuron, while the 3 C subtype genes are expressed by most neurons. Second, using elegant paired electrophysiological recordings, the authors show that in gamma protocadherin knockout cortical slices, the likelihood of two neurons on layers 2/3 being synaptically connected is increased. That suggests that gamma protocadherins generally inhibit synaptic connectivity in the cortex; again, this has been reported previously using morphological assays, but it is helpful to see it confirmed here with physiology. Finally, the authors use an impressive sequential in utero electroporation method to provide evidence that the degree of isoform matching between two neurons negatively regulates their reciprocal synaptic connectivity. These are difficult experiments to do, and while some caveats remain (e.g., lack of demonstration of protein levels in electroporated neurons, lack of resolution of the differences between the present results and those of other papers, a focus on C4 rather than C3 or C5 when considering the highly expressed C-type isoforms), the main result is consistent. Strengths of this manuscript include the impressive methodology and improved demonstration of the previously-reported finding that gamma protocadherins work via homophilic matching to put a brake on synapse formation in the cortex. Because of the unique organization and expression pattern of the gamma protocadherins, it is not likely that these results will be broadly applicable to the general understanding of the role of cell adhesion molecules in synapse development. However, the methodology, which is now better described, should be applicable more broadly and the improved demonstration of the role of gamma protocadherin's negative role in cortical synaptogenesis is helpful in confirming earlier studies. There are several differences between the results here and those of other papers on the cortex, as well as those examining other neuronal populations such as spinal cord. The present findings do not resolve them, but adopting genetic approaches rather than overexpression in the future should help.

---

## [Author Response]

The following is the authors’ response to the original reviews.

**Public Reviews:**

**Reviewer #1 (Public Review):**
The manuscript by Zhu and colleagues aimed to clarify the importance of isoform diversity of PCDHg in establishing cortical synapse specificity. The authors optimized 5' single-cell sequencing to detect cPCDHg isoforms and showed that the pyramidal cells express distinct combinations of PCDHg isoforms. Then, the authors conducted patch-clamp recordings from cortical neurons whose PCDHg diversity was disrupted. In the elegant experiment in Figure 3, the authors demonstrated that the neurons expressing the same sets of cPCDHg isoforms are less likely to form synapses with each other, suggesting that identical cPCDHg isoforms may have a repulsive effect on synapse formation. Importantly, this phenomenon was dependent on the similarity of the isoforms present in neurons but not on the amount of proteins expressed.One of the major concerns in an earlier version was whether PCDHg isoforms, which are expressed at a much lower level than C-type isoforms, have true physiological significance. The authors conducted additional experiments to address this point by using PCDHg cKO and provided convincing data supporting their conclusion. The results from PCDHg C4 overexpression, showing no impact on synaptic connectivity, further clarified the importance of isoforms. I have no further concerns, however, I would like to point out that the evidence for the necessity of the PCDHg isoform is still lacking because most experiments were done by overexpression. It would be helpful for the readers if the authors could add this point to the discussion.

Thank you for the positive feedback on our work. We have now incorporated a discussion of the limitations associated with overexpression.

**Reviewer #2 (Public Review):**
This short manuscript by Zhu et al. describes an investigation into the role of gamma protocadherins in synaptic connectivity in the mouse cerebral cortex. First, the authors conduct a single-cell RNA-seq survey of postnatal day 11 mouse cortical neurons, using an adapted 10X Genomics method to capture the 5' sequences that are necessary to identify individual gamma protocadherin isoforms (all 22 transcripts share the same three 3' "constant" exons, so standard 3'biased methods can't distinguish them). This method adaptation is an advance for examining individual gamma transcripts, and it is helpful to publish the method, the characterization of which is improved in this revised manuscript. The results largely confirm what was known from other approaches, which is that a few of the 19 A and B subtype gamma protocadherins are expressed in an apparently stochastic and combinatorial fashion in each cortical neuron, while the 3 C subtype genes are expressed ubiquitously. Second, using elegant paired electrophysiological recordings, the authors show that in gamma protocadherin cortical slices, the likelihood of two neurons on layers 2/3 being synaptically connected is increased. That suggests that gamma protocadherins generally inhibit synaptic connectivity in the cortex; again, this has been reported previously using morphological assays, but it is important to see it confirmed here with physiology. Finally, the authors use an impressive sequential in utero electroporation method to provide evidence that the degree of isoform matching between two neurons negatively regulates their reciprocal synaptic connectivity. These are difficult experiments to do, and while some caveats remain, the main result is consistent. Strengths include the impressive methodology and improved demonstration of the previously-reported finding that gamma protocadherins work via homophilic matching to put a brake on synapse formation in the cortex. Weaknesses include the writing, which even in the revision fails to completely put the new results in context with prior work, which together has largely shown similar results; a still-incomplete characterization of a new alpha protocadherin KO mouse (a minor point but it should still be addressed); and a lack of demonstration of protein levels in electroporated brains. Because of the unique organization and expression pattern of the gamma protocadherins, it is unlikely that these results will be directly applicable to the broader understanding of the role of cell adhesion molecules in synapse development. However, the methodology, which is now better described, should be applicable more broadly and the improved demonstration of the role of gamma protocadherin's negative role in cortical synaptogenesis is helpful.

Thank you for the positive comments on our work. We have taken your suggestion into account and expanded our discussion to contextualize our research within the broader field of PCDH. Additionally, we have included more data to further illustrate the decrease in αPCDH expression in Pcdha conditional knockout mice. Your feedback has been invaluable in enhancing our manuscript.

**Reviewer #3 (Public Review):**
In this study, Zhu and authors investigate the expression and function of the clustered Protocadherins (cPcdhs) in synaptic connectivity in the mouse cortex. The cPcdhs encode a large family of cadherin-related transmembrane molecules hypothesized to regulate synaptic specificity through combinatorial expression and homophilic binding between neurons expressing matching cPcdh isoforms. But the evidence for combinatorial expression has been limited to a few cell types, and causal functions between cPcdh diversity and wiring specificity have been difficult to test experimentally. This study addresses two important but technically challenging questions in the mouse cortex: (1) Do single neurons in the cortex express different cPcdh isoform combinations? and (2) Does Pcdh isoform diversity or particular combinations among pyramidal neurons influence their connectivity patterns? Focusing on the Pcdh-gamma subcluster of 22 isoforms, the group performed 5'end-directed single-cell RNA sequencing from dissociated postnatal (P11) cortex. To address the functional role of Pcdhg diversity in cortical connectivity, they asked whether the Pcdhgs and isoform matching influence the likelihood of synaptic pairing between 2 nearby pyramidal neurons. They performed simultaneous whole-cell recordings of 6 pyramidal neurons in cortical slices, and measured paired connections by evoked monosynaptic responses. In these experiments, they measured synaptic connectivity between pyramidal neurons lacking the Pcdhgs, or overexpressing dissimilar or matching sets of Pcdhg isoforms introduced by electroporation of plasmids encoding Pcdhg cDNAs.

Overall, the study applies elegant methods that demonstrate that single cortical neurons express different combinations of Pcdh-gamma isoforms, including the upper layer Pyramidal cells that are assayed in paired recordings. The electrophysiology data demonstrate that nearby Pyramidal neurons lacking the entire Pcdhg cluster are more likely to be synaptically connected compared to the control neurons, and that overexpression of matching isoforms between pairs decreases the likelihood to be synaptically connected. These are important and compelling findings that advance the idea that the Pcdhgs are important for cortical synaptic connectivity, and that the repertoire of isoforms expressed by neurons influence their connectivity patterns potentially through a self/nonself discrimination mechanism. However, the findings are limited to probability in connectivity and do they do not support the authors' conclusions that Pcdhg isoforms regulate synaptic specificity, 'by preventing synapse formation with specific cells' or to 'unwanted partners'. Characterizations of the cellular basis of these defects are needed to determine whether they are secondary to other roles in cell positioning, axon/dendrite branching and synaptic pruning, and overall synaptic formation. Claims that Pcdh-alpha and Pcdhg C-type isoforms are not functionally required are premature, due to limitations of the experiments. Moreover, claims that 'similarity level of γPCDH isoforms between neurons regulate the synaptic formation' are not supported due to weak statistical analyses presented in Fig4. The overstatements should be corrected. There was also missed opportunity to clearly discuss these results in the context of other published work, including recent publications focused on the cortex.

Thank you for your feedback on the strengths and weaknesses of our work. In terms of the cellular basis of affected synaptic connectivity caused by γ-PCDH isoforms, we have compared the probability of connectivity for neuronal pairs with similar range of distance. Our findings indicate that the manipulation primarily affects pairs within the 50-150 micrometer range, suggesting that cell positioning might be a critical factor for the impact of γ-PCDH on synapse formation. However, we acknowledge that we couldn't definitively determine whether the negative effect on synaptic connectivity stems directly from impaired synapse formation or indirectly from synaptic pruning or the influence of PCDHγ on axon/dendritic branching. We've added these limitations to our discussion to provide a more comprehensive view of our research. Furthermore, we've adjusted our statements to better reflect the significance of our findings. Your feedback has been instrumental in improving the clarity and depth of our manuscript.

Strengths:The 5' end sequencing with a Pcdhg-amplified library is a technical feat and addresses the pitfall of conventional scRNA-Seq methods due to the identical 3'sequences shared by all Pcdhg isoform and the low abundance of the variable exons. New figures with annotated cell types confirm that several pyramidal and inhibitory cortical subpopulations were captured.

Statistical assessment of co-occurrence of isoform expression within clusters is also a strength.

By establishing the combinatorial expression of Pcdhgs by maturing pyramidal cells, the study further substantiates the 'single neuron combinatorial code for cPcdhs' model. Although combinatorial expression is not universal (ie. serotonergic neurons), there was limited evidence. The findings that individual pyramidal neurons express ~1-3 variable Pcdhg transcripts plus the Ctype transcripts aligns with single RT-PCR studies of single Purkinje cells (Esumi et al 2005; Toyoda et al 2014). They differ from the findings by Lv et al 2022, where C-type expression was lower among pyramidal neurons. OSNs also do not substantially express C-type isoforms (Mountoufaris et al 2017; Kiefer et al 2023). Differences, and the advantages of the 5'end -directed sequencing (vs. SmartSeq) could be raised in the discussion.Simultaneous whole-cell recordings and pairwise comparisons of pyramidal neurons is a technically outstanding approach. They assess the effects of Pcdhg OE isoform on the probability of paired connections.The connectivity assay between nearby pairs proved to be sensitive to quantify differences in probability in Pcdhg-cKO and overexpression mutants. The comparisons of connectivity across vertical vs lateral arrangement are also strengths. Overexpressing identical Pcdhg isoform (whether 1 or 6) reduces the probability of connectivity, but there are caveats to the interpretations (see below).Weaknesses:n earby pairs but are not sufficient evidence for synapse specificity. The cPcdhs play multiple roles in neurite arborization, synaptic density, and cell positioning. Kostadinov 2015 also showed that starburst cells lacking the Pcdhgs maintained increased % connectivity at maturity, suggesting a lack of refinement in the absence of Pcdhgs. The known roles raise questions on how these manipulations might have primary effects in these processes and then subsequently impact the probability of connectivity. Investigations of morphological aspects of pyramidal development would strengthen the study and potentially refine the findings. The authors should more clearly relate their findings to the body of cPcdh studies in the discussion.

Previous studies revealed the adverse effects of γ-PCDHs on dendritic spines, demonstrating that their absence results in increased dendritic spines density, while overexpression leads to a reduction. In our study, we consistently observed that γ-PCDHs exert a negative influence on synaptic connectivity. This consistency strengthens the overall body of evidence in support of the role of γ-PCDHs in synaptic connectivity and dendritic spine regulation. While we have previously mentioned this point in our discussion to highlight the concordance between our findings and prior research, your input is greatly appreciated in reinforcing the scientific context of our work.

Pcdhg cKO-dependent effects on connectivity occur between closely spaced soma (50-100um - Figure 2E), highlighting the importance of spatial arrangement to connectivity (also noted by Tarusawa 2016). Was distance considered for the overexpression (OE) assays, and did the authors note changes in cell distribution which might diminish the connectivity? Recent work by Lv et al 2022 reported that manipulating Pcdhgs influences the dispersion of clonally-related pyramidal neurons, which also impacts the likelihood of connections. Overexpression of Pcdhgc3 increased cell dispersion and decreased the rate of connectivity between pairs. Though these papers are mentioned, they should be discussed in more detail and related to this work.

Our data indicated that variable γ-PCDH isoforms primarily influence synaptic connectivity in neuronal pairs within the 50-150 micrometer range. Notably, as the distance between neurons increases, we observed a corresponding reduction in synaptic connectivity, as illustrated in Figure 2E. We have also included additional discussion regarding potential variances among different C-type isoforms.

Though the authors added suggested citations and improved the contextualization of the study, several statements do not accurately represent the cited literature. It is at the expense of crystalizing the novelty and importance of this present work. For instance, Garrett et al 2012 PMID: 22542181 was the first to describe roles for Pcdhgs in cortical pyramidal cells and dendrite arborization, and that pyramidal cell migration and survival are intact. Line 52 cited Wang et al 2002, but this was limited to gross inspection. Garrett et al is the correct citation for: 'The absence of γ-PCDH does not cause general abnormality in the development of the cerebral cortex, such as cell differentiation, migration, and survival (Wang et al., 2002).' Second, single cell cPcdh diversity is introduced very generally, as though all neuron types are expected to show combinatorial variable expression with ubiquitous C-Type expression. But those initial studies were limited to Purkinje cells (Esumi 2005 and Toyoda 2014). Profiling of serotonergic neurons and OSN reveals different patterns (citations needed for Chen 2017 PMID: 28450636; Mountofaris et al PMID: 2845063; Canzio 2023 PMID: 37347873), raising the idea that cPcdh diversity and ubiquitous Ctype expression is not universal. Thus, the authors missed the opportunity to emphasize the gap regarding cPcdh diversity in the cortex.

We would like to extend our gratitude to the reviewer for pointing out the citation related to the roles of γ-PCDHs in the neocortex. After a thorough review of both papers, Wang et al., 2002 and Garrett et al., 2012, we concur that it would be more appropriate to cite both of these papers here. Your suggestion to underscore the diverse expression patterns of γPCDHs in neocortical neurons is well-received, and we have integrated this aspect of our findings with previous observations into a new paragraph within the discussion section. Your insights have greatly enriched the depth of our paper, and we genuinely appreciate your contribution.

They have not shown rigorously and statistically that the rate of connectivity changes with% isoform matching. In Figure 4D, comparisons of % isoform matching in OE assays show a single statistical comparison between the control and 100% groups, but not between the 0%, 11% and 33% groups. Is there a significant difference between the other groups? Significant differences are claimed in the results section, but statistical tests are not provided. The regression analysis in 4E suggests a correlation between % isoform similarity and connectivity probability, but this is not sound as it is based on a mere 4 data points from 4D. The authors previously explained that they cannot evaluate the variance in these recordings as they must pool data together. However, there should be some treatment of variability, especially given the low baseline rate of connectivity. Or at the very least, they should acknowledge the limitations that prevent them from assessing this relationship. Claims in lines 230+ are not supported: ' Overall, our findings demonstrate a negative correlation between the probability of forming synaptic connections and the similarity level of γPCDH isoforms expressed in neuron pairs (Fig. 4E)".

We employed a bootstrap method to estimate the potential variance in the analysis presented in Fig. 4E. It's important to note that due to methodological limitations, a comprehensive assessment of variance based solely on recordings from a single animal is challenging. As such, we have adjusted our claims to be more aligned with our observations.

Figure 4 provides connectivity probability, but this result might be affected by overall synapse density. Did connection probability change with directionality (e.g between red to green cells, or green to red cells).

As suggested by the reviewer, we have conducted an analysis to assess the directionality of connections under different conditions. This analysis involved comparing the directionalities of connections following the overexpression of six variable isoforms, as depicted in Fig. 3E. Upon examining 33 connected OE-Ctrl pairs following the electroporation of these 6 isoforms, we observed 3 pairs with bidirectional connections, 19 pairs with connections from OE to Ctrl, and 11 with connections from Ctrl to OE. To assess the statistical significance of these observations, we applied a Chi-square test. The results from this analysis indicated that there was no significant difference in the directionality of connections. These findings offer further support for the idea that overexpressing multiple γ-PCDH isoforms within a single neuron might not be sufficient to alter its connections with other neurons.

Generally, the statistical approaches were not sufficiently described in the methods nor in the figure legends, making it difficult to assess the findings. They do not report on how they calculated FDR for connectivity data, when this is typically used for larger multivariate datasets.

We employed the False Discovery Rate (FDR) correction, specifically the BenjaminiHochberg method, to determine which values remained statistically significant. This method is widely accepted and involves inputting all the p-values and the total number, 'n.' Additional details about this correction are now provided in the Method section for clarity.

The possibility that the OE effects are driven by total Pcdhg levels, rather isoform matching, should be examined. As shown by qRT-PCR in Fig. 3, expression of individual isoforms can vary. It is reasonable that protein levels cannot be measured by IHC, although epitope tags could be considered as C-terminal tagging of cPcdhs preserves the function in mice (see Lefebvre 2008). Quantification of constant Pcdhg RNA levels by qRT-PCR or sc-RT-PCR would directly address the potential caveat that OE levels vary with isoform combinations.

Through a series of multiple whole-cell recordings, we examined neuronal pairs within the 0% group, where both neurons exhibited overexpression of different combinations of γPCDH isoforms. What we discovered is that the connectivity level within pairs of neurons where both neurons overexpressed γ-PCDH isoforms, pairs with only one neuron overexpressing these isoforms, and pairs with two control neurons (lacking overexpression) was remarkably similar. However, as we incrementally raised the similarity level between the recorded neurons by increasing the overlap in the combinatorial expression of γ-PCDH isoforms, we observed a gradual decrease in the connectivity probability between these neurons. Notably, the connectivity probability reached its minimum when the recorded cells had the exact same combinatorial expression of γ-PCDH isoforms at the 100% similarity level. These findings suggest that the similarity level between neurons, rather than the absolute expression level of γ-PCDH isoforms, plays a critical role in affecting synapse formation.

-A caveat for the relative plasmid expression quantifications in Figure 3-S1 is that IHC was used to amplify the RFP-tagged isoform, and thus does not likely preserve the relationship between quantities and detection.

We attempted to enhance the mNeongreen signal, known for its exceptional signal-tonoise ratio, by utilizing the 32f6-100 antibody from Chromotek. However, our observations did not reveal any additional cells through immunostaining compared to the images obtained solely based on the mNeongreen signal. This indicates that the application of the available antibody did not yield a significant improvement in cell detection.

It's important to emphasize that if the RFP signal is overvalued, it would result in an increase in both the "red only" and "red in total" categories. However, it's worth noting that the "red only" category is more sensitive to the outcome than the "red in total" category. Therefore, an overvaluation of the RFP signal would lead to an underestimation of the total estimated plasmid content in electroporated neurons. Consequently, this would result in a lower estimate for the proportion of co-expression cells rather than a higher estimate. We have updated the calculation method in the "Estimating the numbers of overexpressed γPCDH isoform" section to reflect these considerations.

Figure 1 didn't change in response to reviews to improve clarity. New panels relating to the scRNASeq analyses were added to supplementary data but many are central and should be included in Figure 1 (ie. S1-Fig6D). In the Results, the authors state that neuronal subpopulations generally show a combinatorial expression of some variable RNA isoforms and near ubiquitous C-type expression. But they only show data for the Layer 2/3 neuron-specific cluster in S1-Fig-6D, and so it is not clear if this pattern applies to other clusters. Fig. S1-5 show a low number of expressed isoforms per cell, but specific descriptions on whether these include C-type isoforms would be helpful. Figure 1F showing isoform profile in all neurons is not particularly meaningful. There is a lot of interest in neuron-type specific differences in cPcdh diversity, and the authors could highlight their data from S1-5 accordingly.

In addition to the layer 2/3 cluster, we observed a diverse combinatorial expression of various variable γ-PCDH isoforms alongside nearly ubiquitous C-type expression in all other clusters of cells. We have now explicitly mentioned this observation in the main text. To underscore this point further, we have included a new figure, Fig. 1-S6, which provides information on the similarity analysis for all other clusters. It's important to note that the data in previous Fig. S1-5 (now renumbered as S1-7) were solely related to "variable" isoforms. We apologize for any confusion and have made this clarification by including it in the title of the figure.

The concept of co-occurrence and results should be explained within the results section, to more clearly relate this concept to data and interpretations. Explanations are now found in the methods, but this did not improve the clarity of this otherwise very interesting aspect of the study.

Thanks for your suggestion. We have incorporated some of the explanations from the methods section into the main text t, mainly for the concept of “co-occurence”.

The claim that C-type Pcdhgs do not functionally influence connectivity is premature. Tests were limited to PcdhgC4, which has unique properties compared to the other 2 C-type isoforms (Garrett et al 2019 PMID: 31877124; Mancia et al PMID: 36778455). The text should be corrected to limit the conclusion to PcdhgC4, and not generally to C-type. The authors should test PcdhgC3 and PcdhgC5 isoforms.

We have changed the claim for PcdhgC4, but not generally for C-type to better reflect our observation.

The group generated a novel conditional Pcdh-alpha mouse allele using CRISPR methods, and state that there were no changes in synaptic connectivity in these Pcdh-alpha mutants. But this claim is premature. The Southern blots validate the targeting of the allele. But further validations are required to establish that this floxed allele can be efficiently recombined, disrupting Pcdha protein levels and function. Pcdha alleles have been validated by western blots and by demonstration of the prominent serotonergic axonal phenotype of Pcdha-KO (ie. Chen 2017 PMID: 28450636; IngEsteves 2018 PMID: 29439167).

We have obtained a new set of qRT-PCR data that confirms the decreased expression of α-PCDH in Pcdha CKO mice. These data have been integrated into Figure 2-S2D.

The Discussion would be strengthened by a deeper discussion of the findings to other cPcdh roles and studies, and of the limitations of the study. The idea that the Pcdhgs are influencing the rate of connectivity through a repulsion mechanism or synaptic formation (ie through negative interactions with synaptic organizers such as Nlgn - Molumby 2018, Steffen 2022) could be presented in a model, and supported by other literature.

I would like to express my sincere appreciation to the reviewer for their invaluable comments and suggestions, which have led to extended discussions within our work. We have incorporated these suggestions into our paper to establish stronger connections between our observations and prior research findings.

**Reviewer #1 (Recommendations For The Authors):**
1. In Figure S6, the authors measured Euclidean distance from the single cell data to take account of the isoform expression levels in explaining diversity. However, it is hard to interpret the data without any control. The authors could measure the same value from a shuffled /randomized dataset for comparison (similarly to Fig 1F).

We understand the reviewer's concern about the significance of the Euclidean distance analysis without an appropriate control. The inclusion of the Euclidean distance metric was initially a response to suggestions from other reviewers who recommended incorporating diverse methods for analyzing expression patterns among neurons.

In response to your valuable feedback, we have taken measures to address these concerns. We have introduced shuffled data for comparison, thus enhancing the meaningful context for interpreting the results derived from the Euclidean distance analysis.

2. The authors need to clarify which cortical regions were used for electrophysiological experiments.

Apologies for any confusion. To clarify, all recordings were conducted on neurons located in layer 2/3 of the neocortex without further discrimination. We have reinstated this information in both the main text and the methods section to ensure its clarity.

**Reviewer #2 (Recommendations For The Authors):**
There are still some issues that must be addressed.1. The references to gamma protocadherin repulsion are not correct in context. A repulsive role of homophilic interaction has been inferred from certain knockout phenotypes in a subset of neurons (not in cortical neurons). However, repulsion has never been shown to follow gamma protocadherin engagement. The authors present no new evidence that their results are attributable to cellular repulsion at nascent synaptic contacts. The mechanism is unknown. The references to repulsion to explain their results should make it clear that this is one possible explanation, but it is not shown. Also some references in the text are not correct. For example, line 63/64: the results of Molumby and Steffen are not involving homophilic adhesion or repulsion, but rather a cis interaction with neuroligins. Those papers should not be discussed as involving repulsion as in the reference to Lefebvre 2012. Also line 268/269 "Together with previous findings (Molumby,,,Tarusawa), our observations solidify repulsion effect of g-PCDH on synapse formation. . .". This is not the case. Neither Molumby nor Tarusawa demonstrated any such repulsion.

Thank you to the reviewer for pointing out the errors in our citations. We have made the necessary corrections to the citations and have also refined the descriptions of our observations to improve clarity and accuracy.

2. The discussion of the results when C4 is overexpressed must also be greatly toned down. C4 is a strange C-type protein--it cannot get to the cell surface alone but relies on other cPCDHs for this, and its primary role is in preventing cell death. It is odd that the authors used this isoform to represent C-types. They should have used C3, which two recent papers showed have specific roles at some synapses (Meltzer et al 2023, Ginty lab) and in dendrite branching (Steffen et al 2023, Weiner lab) , or C5. It is entirely possible that just C4 has no role in synaptic matching--but C3 and C5 might. They should not conclude that the C-types have no such role and only A and B types do. That must be toned down (e.g., line 198/199, line 281).

We acknowledge that using C4 to represent all three C-types (C3, C4, and C5) is not accurate. We have now modified the statement in the main text to rectify this.

3. For the citation of Pcdhg flox/flox mice (line 126), Prasad et al., Development, 2008, Weiner lab, should also be cited as it fully characterized that line that was also used in Lefebvre et al 2008. They were co-published.

Thank you for highlighting the missing citation, and we have now included it in the relevant section.

4. the Pcdh alpha KO Mouse characterization is still insufficient. The authors must show that alpha expression is gone following introduction of Cre, either by RT-PCR using alpha constant domain primers, or an alpha antibody on Western. blot. The southern and off-target sequencing do not confirm that all alpha gene expression is gone.

Thank you for your feedback. We have conducted the qRT-PCR analysis as per your suggestion. The results clearly indicate a substantial reduction in α-PCDH expression within the neocortex of Pcdha cKO mice. We have thoughtfully incorporated this data into the manuscript, and it is visually represented in the new panel of Figure 2-S2D. Your valuable input has contributed to enhancing the quality of our work, and we sincerely appreciate the opportunity to address this important aspect.

5. I do not understand something in Figure 4-S1A. Why with 0% matching is synaptic connectivity so low? This is not the same as in Figure 3E. This has to be explained because it does suggest that overexpression of ANY isoforms can inhibit synapse formation, which is consistent with Molumby 2017, even though this paper says it is not just the levels but the isoform specificity.

The panel of Fig.4-S1A illustrates the connection rate between neurons with the same color (icons in upper left), representing cells that express the same combination of γ-PCDHs (100% of similarity). The X-axis (0%, 11%, 33%, and 100%) reflects the similarity level between the 2 populations of cells (GFP and RFP).

6. There are still issues with the English grammar in the paper. It is not too bad in the main text but someone should re-edit it. However, the figure legends are indeed much worse and truly must be edited professionally before they are acceptable.

We apologize for our English writings in the paper. We have now polished most part of the manuscript, especially the parts for figure legends.

**Reviewer #3 (Recommendations For The Authors):**
This study has many strengths and innovative findings. Most comments above included suggestions to strengthen the paper. The overall message that Pcdhgs influence the rate of synaptic connectivity between nearby cells is compelling. How this Pcdhg-isoform-dependent process could influence synaptic specificity can be explored in a model in the discussion. But this study did not test a role in 'synaptic specificity'; this term should be removed from the title and line 81 in the intro.

Thank you for your invaluable comments aimed at improving our paper. Regarding the title, we believe that "synaptic connectivity" might be a more suitable choice than "synaptic specificity." However, we're open to considering other alternatives as well.

The manuscript and overall quality of the science will be improved by removing those sections that are not adequately investigated (ie.Pcdh-a cKO; PcdhgC4 is assessed but findings can't be extended to other C-type isoforms) and by outlining limitations of the study.

We have modified the related claim mentioned in the main text.

The studies negatively correlating between isoform matching and connectivity are not robust. Additional approaches are needed if the authors want to make this claim.

In Figure 4E, we have implemented a bootstrapping method. Bootstrapping is a statistical technique falling under the broader category of resampling methods. It involves random sampling from the observed data with replacement, enabling the calculation of standard errors, confidence intervals, and supporting hypothesis testing.

Statistical approaches should be described in methods, figure legends.

More information about statistical approaches has been added in the figure legends.

The discussion should elaborate on the limitations of the study, and relate to other studies, including Lv et al 2022.

We have added more discussion to relate our observations to previous findings.